# Implications for Validation of IMERG Satellite Precipitation in a Complex Mountainous Region

Luhan Li [1,2], Xuelong Chen [2,*], Yaoming Ma [2,1,3,4,5,6], Wenqing Zhao [2], Hongchao Zuo [1], Yajing Liu [2], Dianbin Cao [7] and Xin Xu [2]

1. College of Atmospheric Science, Lanzhou University, Lanzhou 730000, China; lilh18@lzu.edu.cn (L.L.); zuohch@lzu.edu.cn (H.Z.)
2. Land-Atmosphere Interaction and Its Climatic Effects Group, State Key Laboratory of Tibetan Plateau Earth System, Environment and Resources (TPESER), Institute of Tibetan Plateau Research, Chinese Academy of Sciences, Beijing 100101, China; ymma@itpcas.ac.cn (Y.M.); zhaowq@itpcas.ac.cn (W.Z.); liuyajing@itpcas.ac.cn (Y.L.); xuxin@itpcas.ac.cn (X.X.)
3. College of Earth and Planetary Sciences, University of Chinese Academy of Sciences, Beijing 100049, China
4. National Observation and Research Station for Qomolongma Special Atmospheric Processes and Environmental Changes, Dingri 858200, China
5. Kathmandu Center of Research and Education, Chinese Academy of Sciences, Beijing 100101, China
6. China-Pakistan Joint Research Center on Earth Sciences, Chinese Academy of Sciences, Islamabad 45320, Pakistan
7. Alpine Paleoecology and Human Adaptation Group, State Key Laboratory of Tibetan Plateau Earth System, Environment and Resources (TPESER), Institute of Tibetan Plateau Research, Chinese Academy of Sciences, Beijing 100101, China; dbcao@itpcas.ac.cn
* Correspondence: x.chen@itpcas.ac.cn

**Abstract:** Satellite-based precipitation retrievals such as the Integrated Multi-satellite Retrievals for Global Precipitation Measurement (IMERG), provide alternative data in mountainous regions. In this study, we evaluated IMERG in the Yarlung Tsangbo Grand Canyon (YGC) using ground observations. It was found that IMERG underestimated the total rainfall primarily due to under-detection of rainfall events, with misses being more prevalent than false alarms. We analyzed the relationships between the probability of detection (POD), false alarm ratio (FAR), bias in detection (BID), and Heidke skill score (HSS) and terrain factors. It was found that the POD decreased with elevation, leading to increased underestimation of rainfall events at higher elevations, and the FAR was higher in valley sites. In terms of the hit events, IMERG overestimated the light rainfall events and underestimated the heavy rainfall events and the negative bias in the hit events decreased with elevation. IMERG could capture the early morning peak precipitation in the YGC region but underestimated the amplitude of the diurnal variation. This bias was inherent at the sensor level, and the Global Precipitation Climatology Center (GPCC) calibration partially improved the underestimation. However, this improvement was not sufficient for the YGC region. This study fills the gap in IMERG validation in a complex mountainous region and has implications for users and developers.

**Keywords:** GPM; Yarlung Tsangbo Grand Canyon; underestimation; validation; diurnal variation; rainfall events under-detection

## 1. Introduction

The Tibetan Plateau (TP), known as the Third Pole [1], is the world's highest plateau. The Asian monsoon passes over the southeastern Tibetan Plateau (SETP) and has an important impact on the climate and environment of the plateau. The Yarlung Tsangbo Grand Canyon (YGC) in the SETP is crucial to the TP region's water cycle and is recognized as one of the largest water vapor canyons [2]. The YGC plays a critical role in transporting water vapor from the South Asian monsoon to the plateau [2]. Thus, examining the water

and energy circulation processes in the YGC region is of great significance due to its strong influence on the TP climate system.

One of the key factors affecting the water and energy cycles is precipitation, for which conventional observations are primarily dependent on rain gauges and ground-based radars. However, rain gauges face challenges in accurately estimating precipitation over large areas with complex terrain. Moreover, in the mountainous regions, the available radars must overcome issues such as beam blockage, anomalous propagation errors, and imprecise backscatter to rain rate relationships [3]. The scarcity of long-term, well-distributed precipitation observations over the complex terrain of the TP highlights the necessity for alternative data sources. Satellite-based precipitation retrievals can fill these data gaps [4].

Currently, global satellite-based rainfall products utilize a combination of microwave, infrared (IR), and integrated microwave and IR observations from multiple satellite missions using a diverse range of merging methods [3]. Early space-based precipitation retrieval efforts concentrated on estimating rainfall through infrared measurements of cloud tops from geosynchronous satellites. Nonetheless, the accuracy was limited due to the indirect association between surface rain rates and cloud-top temperatures [5]. Although infrared sensors on geostationary satellites can provide high temporal resolution precipitation estimates, microwave sensors are preferred for measuring precipitation because their radiative signatures are more closely connected to the precipitating particles. Currently, the most widely used satellite precipitation products (SPPs) include the Tropical Rainfall Measurement Mission (TRMM), Precipitation Estimation from Remotely Sensed Information using Artificial Neural Networks (PERSIANN) [6], Climate Prediction Center morphing technology (CMORPH) [7], Global Satellite Mapping of Precipitation (GSMaP) [8], Climate Hazards Group InfraRed Precipitation with Station data (CHIRPS) [9], Multi-Source Weighted-Ensemble Precipitation (MSWEP) [10], and Global Precipitation Measurement (GPM) Mission [11].

The GPM Mission, as the successor to the TRMM, aims to deliver uniform global precipitation products using a diverse array of microwave sensors within a consistent framework. The GPM Level 3 product, the Integrated Multi-satellite Retrievals for GPM (IMERG), combines intermittent precipitation estimates from all constellation microwave sensors with the more frequent but less accurate IR-based observations from geosynchronous satellites and monthly surface precipitation gauge data. This process creates a consistently calibrated, uniformly gridded global precipitation dataset, as well as relevant error and metadata information [3]. Compared to other datasets, IMERG has superior temporal (0.5 h) and spatial ($0.1° \times 0.1°$) resolutions, providing a more refined data source for studying climate change and enhancing weather forecasts on the TP. Consequently, the performance of IMERG in the plateau region has attracted considerable attention. Previous evaluations of IMERG over the TP have focused on whether IMERG serves as a suitable successor to the TRMM Multi-satellite Precipitation Analysis (TMPA) and comparing its performance to other precipitation products for the plateau region.

Throughout the TP, compared to the TMPA, IMERG inherits the spatial precipitation pattern and the ability to capture the northward progression of the Indian Monsoon lifecycle over the TP from its predecessor [12]. IMERG has a higher similarity to ground-based observations, enhanced detection capabilities, and reduced errors, and it mitigates the TMPA's overestimation over the TP. Simultaneously, IMERG improves the detection of light rainfall events and the potential for solid precipitation [12,13]. This is closely related to the increased radar frequency and the inclusion of the imager's high-frequency channel in the GPM platform [14]. Notably, IMERG can adequately depict diurnal precipitation characteristics [2]; however, discrepancies arise when describing the timing of the maximum intensity, and overestimation of the maximum precipitation rate occurs [13]. Compared to other satellite products, IMERG's performance is more prominent, which is mainly reflected in its ability to detect precipitation and non-precipitation events [15], precipitation data correction ability [16], and hydrological modeling potential [17,18].

Nevertheless, IMERG still has some shortcomings. First, IMERG overestimates the precipitation frequency in multiple regions of the TP, such as the eastern edge of the TP and the Sichuan Basin [17], the southeastern area during the rainy season [18], and the Yarlung Tsangbo River region [19]. Second, IMERG's detection capabilities are insufficient at high altitudes [12,20,21]. Third, IMERG overestimates light rainfall events and underestimates moderate to heavy rainfall events [19]. This phenomenon is not only present in IMERG but is also a common issue with most satellite products [15,20,22].

Existing research has acknowledged the advantages of IMERG in the TP region while also pointing out several remaining issues. However, the current evaluation of IMERG in the TP region still has the following shortcomings.

- Spatial scale: Previous studies have mainly focused on the entire plateau or river basin scales, i.e., research on the Yarlung Tsangbo River [18,22]. Due to the limited number of ground observation stations in the downstream region of the Yarlung Tsangbo River, satellite precipitation studies have mainly concentrated on the upstream region [20]. Our study area, the YGC, is the downstream region of the Yarlung Tsangbo River and is an important region for water vapor transport on the SETP. However, due to its remote location and sparse observation sites, evaluation in this area has not yet received sufficient attention. Moreover, the microclimate characteristics and ecosystems in the YGC differ significantly from those in other basins along the Yarlung Tsangbo River, warranting further investigation. Therefore, we conducted a detailed assessment of IMERG at the grid scale specifically for the YGC region.

- Temporal scale: Existing IMERG validation studies in the TP region have mainly focused on the daily, monthly, seasonal, and annual scales, and only a few studies explored the hourly scale and diurnal variation characteristics [12,13]. However, the half-hourly temporal scale of IMERG is crucial for studying mountain areas such as the YGC, which experience high variations in rainfall frequency [2]. The rain gauge data used in this study were obtained from the Second Tibetan Plateau Scientific Expedition and Research Program (STEP) titled Investigation of the Water Vapor Channel of the Yarlung Tsangbo Grand Canyon (INVC) [23]. The ground-based precipitation observations are available at half-hourly scales, allowing our assessment to be refined to sub-hourly scales.

- Elevation and terrain factors: In the TP region, the accuracy of IMERG precipitation products varies with elevation [17]. Ma et al. [24] reported that IMERG performs better at elevations of 3000–4000 m, while Zhang et al. [13] found differences in the correlation coefficients and relative biases between IMERG and observations at elevations of around 3500 m. Some studies have argued that IMERG performs poorly at high elevations [12,20,21], while others have reported the opposite view [17]. This phenomenon is due to the varying accuracy of the same product in the southern, southeastern, central-eastern, and northeastern TP regions [15]. Moreover, the TP region has complex and heterogeneous terrain with strong local influencing factors. The relationship between IMERG and the terrain in existing studies remains unclear. However, elevation and terrain are the key factors affecting IMERG detection. Therefore, in this study, we also focused on exploring how the elevation and terrain in the YGC region affect the consistency between IMERG and ground-based observations.

- Instrument-level assessment: Tan et al. [5] pointed out that identifying the sources of satellite uncertainties during satellite evaluation can promote satellite development and improvement. Consequently, some studies have investigated the integration process of IMERG and conducted evaluations from the instrument and algorithm perspectives [5]. Currently, few assessments have been conducted from this perspective. In this study, we evaluated the diurnal and seasonal variations derived from different IMERG sensors, which will help IMERG users and developers to better understand the sources of the uncertainties in the IMERG product for the YGC region.

In this study, based on the aforementioned aspects, we utilized ground observation data for the YGC region of the SETP to evaluate IMERG's half-hourly and daily scale data.

The evaluation encompasses precipitation event detection, rainfall amount estimation, and diurnal and seasonal variation reproduction capabilities. Additionally, we investigated the influences of altitude and topography factors on IMERG's detection performance and quantified the product's uncertainties from the instrument perspective. This effectively fills the gap in IMERG validation for the Yarlung Tsangbo River Basin, improves our understanding of the impact of the South Asian Monsoon on the TP's water cycle, and has significant implications for satellite precipitation users and algorithm developers.

The rest of this article is organized as follows. Section 2 describes the study area and the datasets and methods used in this study. Section 3 presents the evaluation results for IMERG against the rain gauge data at half-hourly and daily time scales, including rainfall events, rainfall amounts, different rainfall intensities, daily cycles, seasonal variation, and sensor level performance, as well as an analysis of the influence of the topographic factors on IMERG's performance. The discussion and the conclusions are presented in Sections 4 and 5.

## 2. Datasets and Methods

### 2.1. Study Area and Datasets

The Yarlung Tsangbo River flows from west to east through the southern TP, tracing a dramatic bend around Mount Namcha Barwa, the highest peak in the eastern Himalayas. After passing through Motog County and finally crossing into India, it forms one of the world's deepest canyons [2]. The YGC is approximately 496.3 km long and has a maximum depth of 5382 m [25]. Positioned between the eastern Himalayan Mountains and the Hengduan Mountains, it forms a concave ridge with two windward ridge arms shaping a funnel-like valley (Figure 1a). Its north-south orientation amplifies a channel effect. The warm and humid air from the Indian Ocean goes up the Brahmaputra River, passes through the lower valley of the Yarlung Tsangbo River, and enters into the TP [25,26]. The canyon's influence on the TP's climate predominantly manifests in precipitation patterns. A moisture-rich "wet tongue" extends into the plateau following the canyon's path, resulting in precipitation rates nearly double those of similar latitude plateau peripheries. For the beginning of the rainy season, the plateau is 1–2 months earlier than other areas at the same latitude [25]. Moreover, the channel's favorable hydrothermal conditions create the northern hemisphere's northernmost tropical mountainous environment in the valley. Here, numerous tropical species exceed their typical latitudinal or altitudinal ranges, reaching their distribution's furthest northern or highest limits. Thus, the YGC mediates and impacts the Himalayas' geography, landscape, glaciers, and geological hazards, serving as a pivotal zone for studying climate, topography, and ecosystem interactions.

The study area is situated at 94–96°E, 29–31°N (Figure 1b). Eighteen observation sites (blue dots in Figure 1b, hereafter referred to as INVC-RG) were installed in the YGC, supported by the INVC program [23]. The INVC established these observation stations at various altitudes ranging from 511 to 3328 m [27]. These stations not only cover a significant portion of the YGC but also extend the coverage of the observation area, such as Pailong and Danka, in Yigong Zangbo and Palong Zangbo Valley. This expanded data coverage provides more possibilities for analyzing and modeling dynamic precipitation variations. Furthermore, we incorporated data from four observation sites (black triangles in Figure 1b, hereafter referred to as CMA-RG) managed by the China Meteorological Administration (CMA). Detailed information about these twenty-two observation sites is presented in Table 1.

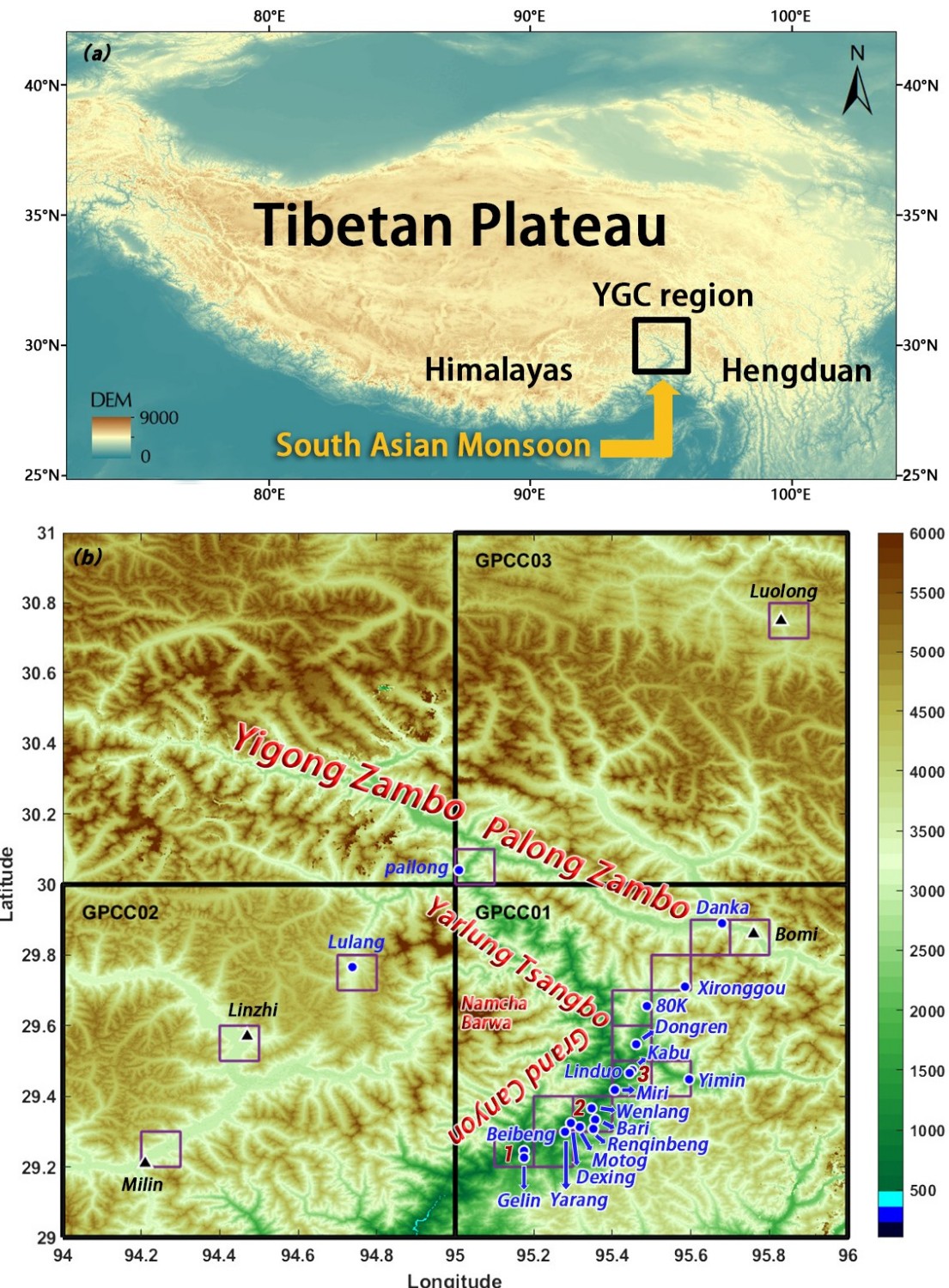

**Figure 1.** (**a**) Locations of the Yarlung Tsangbo Grand Canyon (YGC), the Himalayas, and the Hengduan Mountains on the Tibetan Plateau (TP) and DEM (shaded; unit: m) of the TP. (**b**) DEM of the YGC region (shaded; unit: m) and the locations of 22 rain gauge sites. The blue dots denote the rain gauge sites (from Chen et al. [2]), and black triangles represent the national meteorological stations. The small purple boxes are the IMERG 0.1° × 0.1° grids. The IMERG grids with multiple RGs (IMERG01, IMERG02, and IMERG03) are marked with red numbers (1, 2, 3). The big black boxes denote the GPCC 1° × 1° grids, which are distinguished by GPCC01, GPCC02, and GPCC03. Locations of Yigong Zambo, Palong Zambo, YGC, and the Mount Namcha Barwa are marked in red font.

**Table 1.** Rain gauge sites in the YGC region.

| No. | Site | IMERG GRID | GPCC GRID | Lat. | Lon. | Elevation (m) | Relative Elevation (m) | Analysis Period |
|---|---|---|---|---|---|---|---|---|
| 1 | Beibeng | IMERG01 | GPCC01 | 29.245 | 95.175 | 853 | 243 | Jan. 2019–Sep. 2021 |
| 2 | Gelin | IMERG01 | GPCC01 | 29.226 | 95.175 | 1789 | 1179 | Nov. 2020–Sep. 2021 |
| 3 | Yarang | | GPCC01 | 29.299 | 95.280 | 757 | 141 | Jan. 2019–Sep. 2021 |
| 4 | Dexing | | GPCC01 | 29.324 | 95.294 | 737 | 85 | Jul. 2020–Sep. 2021 |
| 5 | Wenlang | IMERG02 | GPCC01 | 29.366 | 95.347 | 851 | 168 | Jul. 2020–Sep. 2021 |
| 6 | Motog | IMERG02 | GPCC01 | 29.313 | 95.317 | 1300 | 617 | Jan. 2019–Oct. 2020 |
| 7 | Bari | IMERG02 | GPCC01 | 29.334 | 95.357 | 1700 | 1017 | Oct. 2020–Sep. 2021 |
| 8 | Renqinbeng | IMERG02 | GPCC01 | 29.308 | 95.352 | 2058 | 1375 | Aug. 2020–Sep. 2021 |
| 9 | Miri | IMERG03 | GPCC01 | 29.418 | 95.406 | 832 | 129 | Jan. 2019–Sep. 2021 |
| 10 | Linduo | IMERG03 | GPCC01 | 29.465 | 95.443 | 840 | 137 | Oct. 2020–Sep. 2021 |
| 11 | Kabu | IMERG03 | GPCC01 | 29.473 | 95.450 | 1425 | 722 | Jan. 2019–Sep. 2021 |
| 12 | Yimin | | GPCC01 | 29.447 | 95.595 | 1751 | 424 | Jul. 2020–Sep. 2021 |
| 13 | Dongren | | GPCC01 | 29.547 | 95.461 | 1185 | 416 | Jan. 2019–Sep. 2021 |
| 14 | 80K | | GPCC01 | 29.655 | 95.488 | 2109 | 895 | Jan. 2019–Sep. 2021 |
| 15 | Xironggou | | GPCC01 | 29.710 | 95.585 | 2786 | 265 | Jan. 2019–Aug. 2021 |
| 16 | Danka | | GPCC01 | 29.890 | 95.680 | 2700 | 65 | Jan. 2019–Sep. 2021 |
| 17 | Bomi | | GPCC01 | 29.860 | 95.760 | 2749 | 110 | Jan. 2018–Sep. 2021 |
| 18 | Milin | | GPCC02 | 29.210 | 94.210 | 2952 | 92 | Jan. 2018–Sep. 2021 |
| 19 | Linzhi | | GPCC02 | 29.570 | 94.470 | 3001 | 102 | Jan. 2018–Sep. 2021 |
| 20 | Lulang | | GPCC02 | 29.766 | 94.738 | 3328 | 146 | Jan. 2019–Sep. 2021 |
| 21 | Pailong | | GPCC03 | 30.041 | 95.010 | 2081 | 197 | Jan. 2019–Sep. 2021 |
| 22 | Luolong | | GPCC03 | 30.750 | 95.830 | 3640 | 81 | Jan. 2018–Sep. 2021 |

Notation: The sites are numbered from south to north.

The ground-based precipitation data utilized in this study were primarily derived from rain gauges (hereafter referred to as RGs) with the following two main sources. The INVC employs tipping-bucket RGs with a minimum detection unit of 0.2 mm/tip. The data from INVC-RGs were used to analyze the half-hourly precipitation, daily precipitation, diurnal variation of the precipitation (DVP), and seasonal variation. It can be downloaded from the website: http://data.tpdc.ac.cn/zh-hans/disallow/e68f1de1-3a13-4ae1-90e0-9 e3a3f57f912/, accessed on 26 February 2023. The data from CMA-RGs were used to analyze the daily precipitation and seasonal variation, which can be downloaded from http://www.cma.gov.cn, accessed on 27 January 2023.

The 22 stations utilized in this study are independent of the Global Precipitation Climatology Center (GPCC) observation network [28]. In terms of quality control, the physically incorrect values were removed, and apparent erroneous station data were eliminated through comparison with neighboring RGs. Due to the inherent deformation and calibration algorithms, IMERG inevitably generates a substantial amount of very small precipitation values, which are commonly considered to be noise [29]. The precipitation/no-precipitation thresholds for half-hourly and daily data were set to 0.2 mm/0.5 h and 0.2 mm/d. Events below these thresholds were classified as no-precipitation events.

The satellite data used in this study were obtained from the IMERG-Final product (V06 version) with a spatial resolution of $0.1° × 0.1°$. The GPM core observatory serves as a calibration and evaluation tool, integrating all passive microwave (PMW) and IR-based precipitation products into IMERG. Consequently, the IMERG data originates from two sources: (1) the integration of inter-calibrated microwave precipitation estimates from the GPM microwave imager (GMI), TRMM microwave imager (TMI), and all constellation partner sensors; and (2) when low Earth orbit (LEO)-PMW data are sparse, Kalman filter-generated geosynchronous Earth orbit (GEO)-IR precipitation estimates. Additionally, gauge-based analysis was used to provide crucial regionalization and bias correction of the satellite estimates [14].

IMERG has three versions: the early multi-satellite product (IMERG-E, available approximately 4 h after observation), the late multi-satellite product (IMERG-L, available

approximately 14 h after observation), and the final satellite measuring instrument product (IMERG-F, available approximately 3.5 months after observation). Lei et al. [17] compared the three versions in the eastern TP region and found that IMERG-F is in better agreement with gauge observations. For all operational versions, the complete IMERG products encompass calibration against monthly precipitation analyses. However, the distinction lies in the calibration methodology: climatological calibration is applied to the early and late products, while monthly calibration is employed for the final product. Consequently, the climatological coefficients of the final product vary according to the month and location and are recommended for general applications [14]. Due to the intricate terrain of the YGC region, acquiring foundational data poses significant challenges. In the analysis and verification of the YGC area, there is a shortage of high spatiotemporal resolution observational data. IMERG-F's monthly gauge calibration, in comparison to the IMERG-E and IMERG-L versions, is notably more localized. This aspect provides us with increased potential for the validation of subsequent meteorological simulations. Therefore, the objective of this study is to investigate the role of calibration within IMERG-F and to assess the usability of the final product. This exploration aims to furnish more sound data sources for future research. Therefore, the IMERG-F product (hereinafter referred to as IMERG) was applied in this study.

In the half-hourly analysis, the primary data used were IMERG's half-hourly precipitationCal, the multi-satellite precipitation estimate adjusted using the GPCC meteorological gauge data every month. The precipitationCal data is also the officially recommended data for general use. Moreover, in the DVP analysis, the following data were also used: (1) precipitationUncal (hereinafter referred to as Uncal), which is the multi-satellite precipitation estimate without GPCC gauge calibration; (2) HQprecipitation (hereinafter referred to as HQ), which is the merged microwave-only precipitation estimate; and (3) IRprecipitation (hereinafter referred to as IR), which is the IR precipitation estimate [14]. IMERG's one-day and monthly precipitation products were utilized for the daily and seasonal analysis. The IMERG product can be downloaded from https://disc.gsfc.nasa.gov/, accessed on 13 December 2022, and the GPCC product can be downloaded from https://opendata.dwd.de/climate_environment/GPCC/html/download_gate.html, accessed on 8 November 2022.

### 2.2. Performance Analysis Methods

In order to analyze the performance of the IMERG, different evaluation metrics were used, two terrain indices were identified, and the effect of terrain on the detection capability of the IMERG was studied.

The complex topography of the TP significantly influences the precipitation variations at observation stations due to elevation and terrain undulation [30–32]. In our investigation, we explored the impact of topographical factors on IMERG's detection in the YGC region by employing site elevation and relative elevation above the valley bottom. The elevations for the INVC-RGs were sourced from global positioning system measurements, while those for the CMA-RGs were defined by the CMA. The relative elevation was defined as the site elevation minus the minimum digital elevation model (DEM) value of the IMERG grid corresponding to the site's location, which indicates the relative position of the site in the IMERG grid where it is located. Lower relative elevation means the site is closer to the valley. The DEM data were extracted from the ASTER GDEM dataset, featuring a spatial resolution of 30 m (1 arc-second). The elevations and the relative elevations with sites are shown in Figure 2.

For the comparison of the satellite and ground-based precipitation observations, we employed the method by selecting the IMERG grid cell closest to the RG station and directly comparing the ground-based precipitation measurements with the corresponding IMERG grid values [33]. Notably, when an IMERG grid cell contained multiple RG stations, we refrained from spatially averaging the RG data [29].

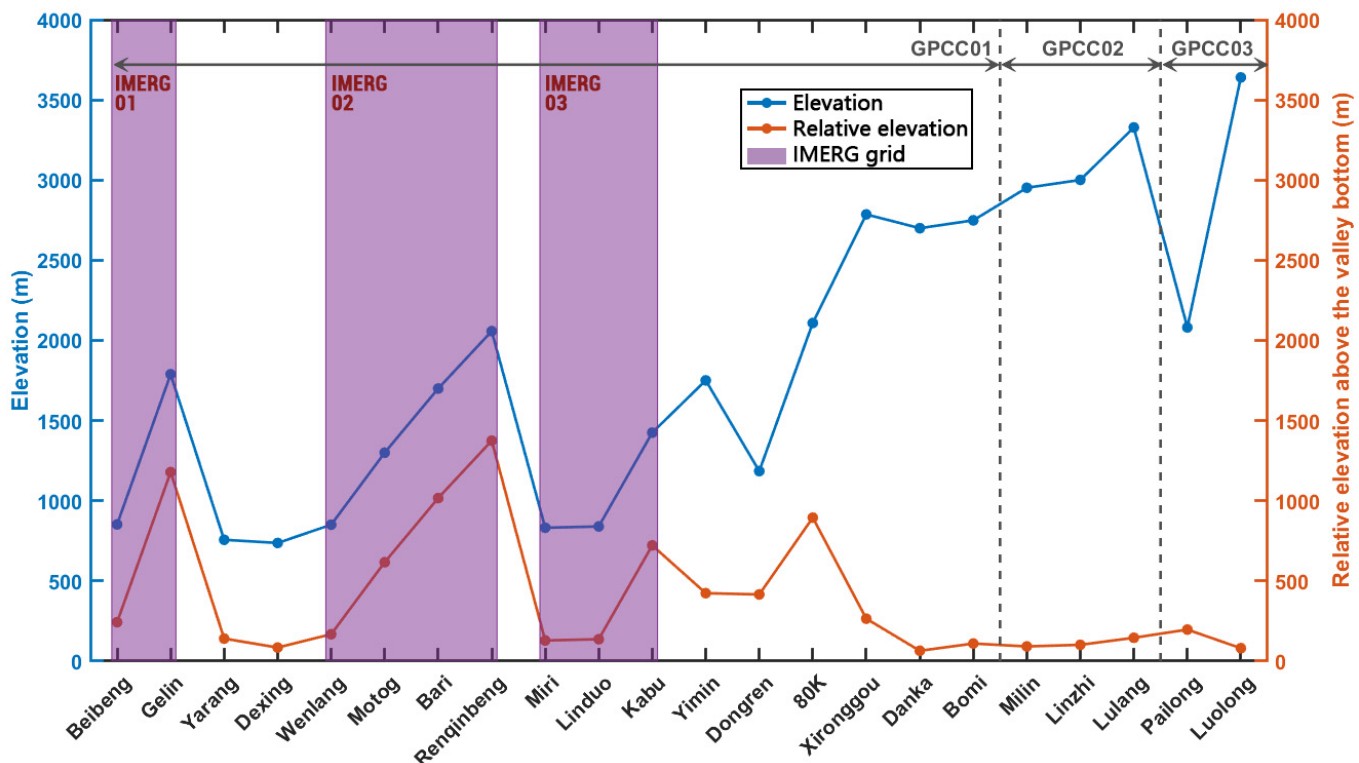

**Figure 2.** The elevations (unit: m) and the relative elevations (unit: m) of 22 rain gauge sites. The purple shading denotes the IMERG01, IMERG02, and IMERG03 grids, and the double arrows demonstrate the range covered by GPCC01, GPCC02, and GPCC03 grids.

In this study, we employed the metrics proposed by Tang et al. [34] and Tan et al. [5] to evaluate the IMERG products. Our approach involved calculating contingency tables that included hits (H), misses (M), false alarms (F), and correct negatives (C) and comparing RG measurements with IMERG rainfall estimates. A hit was recorded when both the RG and IMERG indicated rainfall; a miss was recorded when the RG registered rainfall but the IMERG did not; a false alarm was recorded when IMERG identified rainfall but the RG did not; and a correct negative was recorded when neither the RG nor IMERG reported rainfall. We classified an instance as "raining" when the rain rate reached at least 0.2 mm for half-hourly and daily rainfall. The formulas of the statistical metrics are listed in Table 2.

**Table 2.** List of the statistical metrics used for evaluating IMERG.

| Statistic Metrics | Formula | Perfect Score |
|---|---|---|
| Probability of detection (*POD*) | $POD = \frac{H}{H+M}$ | 1 |
| False alarm ratio (*FAR*) | $FAR = \frac{F}{H+F}$ | 0 |
| Bias in detection (*BID*) | $BID = \frac{H+F}{H+M}$ | 1 |
| Heidke skill score (*HSS*) | $HSS = \frac{H+C-E}{N-E}$ | 1 |
| Expected number of instances (*E*) | $E = \frac{1}{N}[(H+M)(H+F) + (C+M)(C+F)]$ | —— |
| Normalized mean error (*NME*) | $NME = \frac{\frac{1}{n}\sum_i (y_i - x_i)}{\frac{1}{n}\sum_i x_i}$ | 0 |
| Normalized mean absolute error (*NMAE*) | $NMAE = \frac{\frac{1}{n}\sum_i |y_i - x_i|}{\frac{1}{n}\sum_i x_i}$ | 0 |
| Correlation coefficient (*CC*) | $CC = \frac{\sum_{i=1}^{n}(x_i - \bar{x})(y_i - \bar{y})}{\sqrt{\sum_{i=1}^{n}(x_i - \bar{x})^2}\sqrt{\sum_{i=1}^{n}(y_i - \bar{y})^2}}$ | 1 |

Notation: *H*, hits; *M*, misses; *F*, false alarms; *C*, correct negatives; *N*, the sample size; *E*, the expected number of instances that can be correctly identified based solely on random chance; $x_i$ and $y_i$ represent the rain rates for the RG and IMERG, respectively, and n signifies the number of hits. "——" indicates no perfect score.

The probability of detection (POD) measures the proportion of actual rainfall events correctly detected by the estimate, the false alarm ratio (FAR) quantifies the ratio of incorrectly estimated rainfall events, the bias in detection (BID) assesses the tendency of the IMERG estimation to overestimate (BID > 1) or underestimate (BID < 1) the number of rainfall events, and the Heidke skill score (HSS) provides a comprehensive assessment that whether the IMERG estimate performs worse (HSS < 0) or better (HSS > 0) than random chance relative to the RG. After evaluating the rainfall events, we concentrated on the hits and analyzed the rain rates within three conventional measures: the Normalized Mean Error (NME), which evaluates the degree of systematic error in the estimation method; the Normalized Mean Absolute Error (NMAE), which assesses the level of random error; and the Pearson Correlation Coefficient (CC), which quantifies the linear agreement between the estimated rainfall and the reference data.

Regarding the precipitation intensity, we adopted the percentile method based on the approaches of Sharifi et al. [35] and Yu et al. [36]. The half-hourly and daily precipitation samples from RGs were arranged in ascending order. The 50th, 70th, 90th, and 98th percentiles of the ground-based observation data were set as the thresholds for light, moderate, heavy, and extreme precipitation, respectively. The classification of the rainfall intensities was as follows: (1) half-hourly precipitation: light rainfall 0.2–0.4 mm/0.5 h, moderate rainfall 0.4–0.8 mm/0.5 h, heavy rainfall 0.8–1.6 mm/0.5 h, rainstorm 1.6–3.2 mm/0.5 h, extreme rainfall > 3.2 mm/0.5 h; (2) daily precipitation: light rainfall 0.2–3.8 mm/d, moderate rainfall 3.8–8.4 mm/d, heavy rainfall 8.4–20.8 mm/d, rainstorm 20.8–40.2 mm/d, extreme rainfall > 40.2 mm/d.

## 3. Results

### 3.1. Assessment of Precipitation Events

First, we assessed the capability of IMERG to accurately determine the occurrence of precipitation at each RG. Figure 3a presents an evaluation of the half-hourly precipitation events based on 18 INVC-RGs. The statistical results indicate that the half-hourly precipitation events at each site in the YGC region accounted for 6.5% to 22.2% of all of the available data. The hit rate of IMERG ranged from 1.4% to 5.5% for precipitation events and from 75.4% to 90.3% for non-precipitation events. The errors in IMERG's rainfall detection included misses and false alarms, accounting for 4.6–18.9% and 2.1–4.1% of the total events, respectively. For every site within the YGC region, the proportion of misses by IMERG exceeded that of the false alarms. The miss-to-false ratio was the lowest at Lulang (approximately 1.4) and highest at Xironggou (up to 9). This suggests that despite a certain degree of compensation between misses and false alarms, significant under-detection errors persist in IMERG.

Figure 3b assesses the ability of IMERG to capture daily precipitation events observed by 18 INVC-RGs and four CMA-RGs. The daily precipitation is accumulated from half-hourly data, which increases the probability of the precipitation per unit of time, thus increasing the proportion of precipitation events, decreasing the proportion of non-precipitation events, and enhancing the likelihood of IMERG detecting rainfall. By comparing Figure 3a,b, it can be seen that as the timescale increases, the proportions of misses, hits, and false alarms all increase significantly, but a predominance of misses and the proportion of hits is slightly larger than that of false alarms.

Based on the aforementioned evaluation indicators, the four skill scores of each RG were further calculated (Table 3). IMERG detects 10–35% of the half-hourly rainfall events, while on a daily scale, the rate increases to 21–34%. This improvement mirrors the overall trend of the POD, but not all of the stations exhibit this behavior. Half-hourly POD is higher than daily POD to the south of Dongren (Site No. 1–13); however to the north of 80K, daily POD is higher than half-hourly POD (Site No. 14–16, 20, 21), thereby revealing a noticeable north-south differentiation, which is consistent with the spatial correlation analysis of rainfall reported by Chen et al. [2].

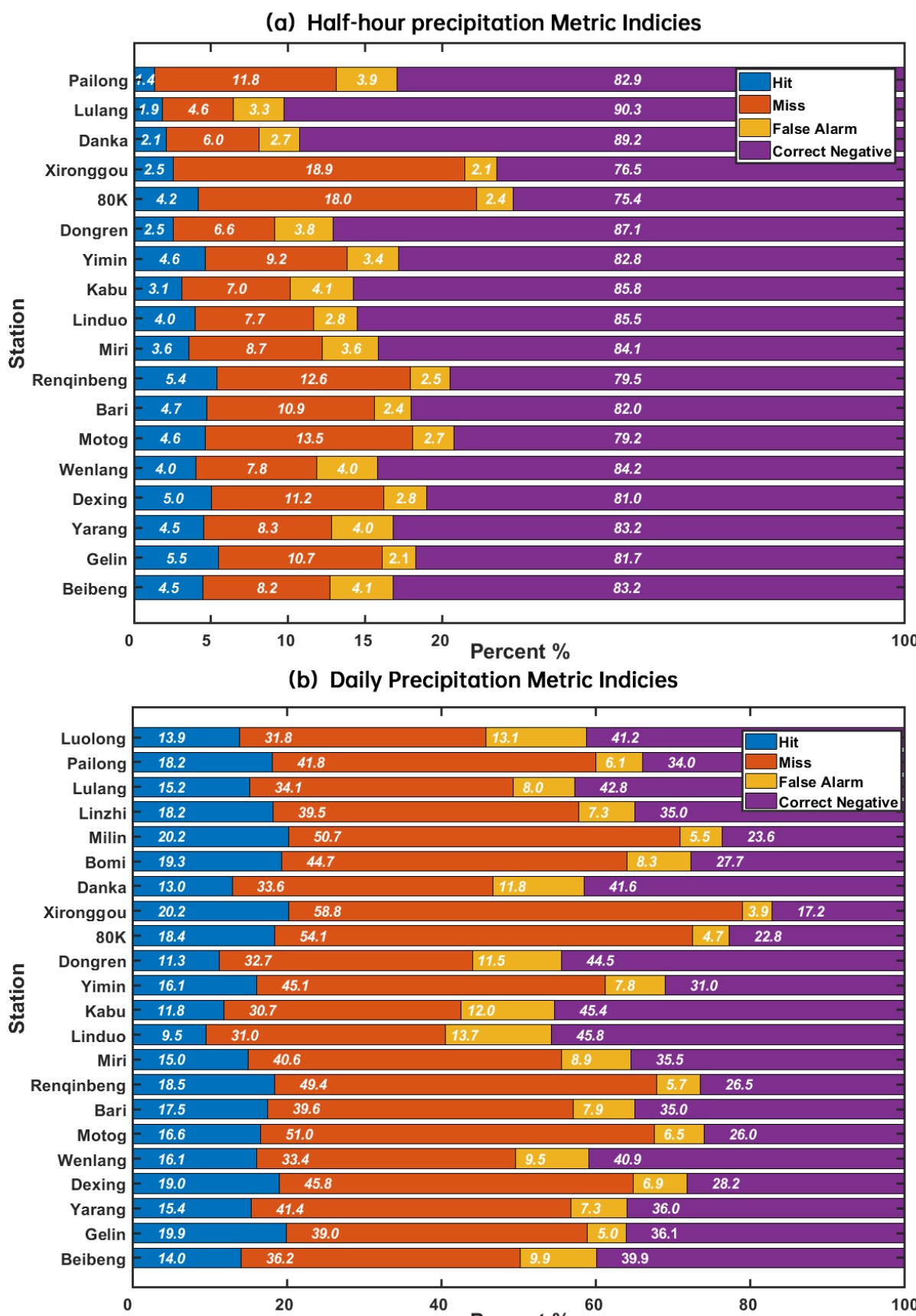

**Figure 3.** Hits, misses, false alarms, and correct negatives for each site on the (**a**) half-hourly and (**b**) daily temporal scales. The white numbers indicate the percentages.

**Table 3.** Probability of detection (POD), false alarm ratio (FAR), bias in detection (BID), and Heidke skill score (HSS) on the half-hourly and daily scales for IMERG.

| No. | Site | IMERG GRID | GPCC GRID | Half-Hourly | | | | Daily | | | |
|-----|------|-----------|-----------|-----|-----|-----|-----|-----|-----|-----|-----|
| | | | | POD | FAR | BID | HSS | POD | FAR | BID | HSS |
| 1 | Beibeng | IMERG01 | GPCC01 | 0.35 | 0.48 | 0.67 | 0.35 | 0.28 | 0.41 | 0.48 | 0.08 |
| 2 | Gelin | IMERG01 | GPCC01 | 0.34 | 0.28 | 0.47 | 0.4 | 0.34 | 0.2 | 0.42 | 0.19 |
| 3 | Yarang | | GPCC01 | 0.35 | 0.47 | 0.66 | 0.36 | 0.27 | 0.32 | 0.4 | 0.09 |
| 4 | Dexing | | GPCC01 | 0.31 | 0.35 | 0.48 | 0.35 | 0.29 | 0.27 | 0.4 | 0.08 |
| 5 | Wenlang | IMERG02 | GPCC01 | 0.34 | 0.5 | 0.68 | 0.34 | 0.33 | 0.37 | 0.52 | 0.14 |
| 6 | Motog | IMERG02 | GPCC01 | 0.26 | 0.37 | 0.4 | 0.29 | 0.25 | 0.28 | 0.34 | 0.03 |
| 7 | Bari | IMERG02 | GPCC01 | 0.3 | 0.33 | 0.46 | 0.35 | 0.31 | 0.31 | 0.45 | 0.11 |
| 8 | Renqinbeng | IMERG02 | GPCC01 | 0.3 | 0.32 | 0.44 | 0.34 | 0.27 | 0.23 | 0.36 | 0.07 |
| 9 | Miri | IMERG03 | GPCC01 | 0.29 | 0.51 | 0.59 | 0.3 | 0.27 | 0.37 | 0.43 | 0.06 |
| 10 | Linduo | IMERG03 | GPCC01 | 0.34 | 0.42 | 0.58 | 0.38 | 0.23 | 0.59 | 0.57 | 0 |
| 11 | Kabu | IMERG03 | GPCC01 | 0.31 | 0.57 | 0.71 | 0.3 | 0.28 | 0.5 | 0.56 | 0.07 |
| 12 | Yimin | | GPCC01 | 0.34 | 0.42 | 0.58 | 0.36 | 0.26 | 0.33 | 0.39 | 0.05 |
| 13 | Dongren | | GPCC01 | 0.28 | 0.6 | 0.69 | 0.27 | 0.26 | 0.51 | 0.52 | 0.05 |
| 14 | 80K | | GPCC01 | 0.19 | 0.36 | 0.29 | 0.21 | 0.25 | 0.2 | 0.32 | 0.05 |
| 15 | Xironggou | | GPCC01 | 0.12 | 0.45 | 0.21 | 0.13 | 0.26 | 0.16 | 0.3 | 0.04 |
| 16 | Danka | | GPCC01 | 0.26 | 0.56 | 0.58 | 0.28 | 0.28 | 0.48 | 0.53 | 0.06 |
| 17 | Bomi | | GPCC01 | —— | —— | —— | —— | 0.3 | 0.3 | 0.43 | 0.06 |
| 18 | Milin | | GPCC02 | —— | —— | —— | —— | 0.29 | 0.21 | 0.36 | 0.07 |
| 19 | Linzhi | | GPCC02 | —— | —— | —— | —— | 0.32 | 0.29 | 0.44 | 0.13 |
| 20 | Lulang | | GPCC02 | 0.29 | 0.64 | 0.8 | 0.28 | 0.31 | 0.34 | 0.47 | 0.15 |
| 21 | Pailong | | GPCC03 | 0.1 | 0.74 | 0.4 | 0.08 | 0.3 | 0.25 | 0.4 | 0.13 |
| 22 | Luolong | | GPCC03 | —— | —— | —— | —— | 0.3 | 0.49 | 0.59 | 0.06 |

Notation: The sites are numbered from south to north. "——" indicates no data.

As the denominator of the POD is determined by both hits and misses, a high proportion of misses within a day can still cause a decrease in the POD score of the daily rainfall, even if IMERG hits a certain number of events within the half-hourly rainfall.

For IMERG's precipitation estimates, the FAR is 28–74% for half-hourly rainfall and approximately 21–59% for daily rainfall (Table 3). Nearly all of the sites exhibit a decrease in the FAR as the temporal scale increases. It is noteworthy that when multiple RGs are located within the same IMERG grid, the satellite estimates have a relatively higher FAR in lower altitude areas. In the YGC area, the maximum yearly rainfall difference within a single grid can reach up to 536 mm [2], indicating significant spatial variability at sub-grid scales. This variability underscores the challenges posed by the low site density and representativeness in the validation of IMERG. IMERG consistently underestimates the occurrence of rainfall across the entire YGC region (BID < 1), corresponding with a high miss ratio. This is a common phenomenon in orographic precipitation [37,38]. Overall, IMERG's estimates have a lower total hit ratio for daily rainfall events, and its comprehensive ability, measured by the HSS, is less than ideal. Therefore, when using IMERG to determine the occurrence of daily rainfall events, a cautious approach should be adopted for the YGC region.

### 3.2. Assessment of Rainfall Quantification

To delve deeper into the system's performance in quantifying rainfall, we will focus on rainfall events correctly hit by IMERG and assess the rainfall amounts (Table 4). The NME and NMAE are used to represent the systematic and random errors of IMERG relative to the RG data, respectively, while the CC is used to measure their concurrence.

**Table 4.** Evaluation of half-hourly and accumulated daily precipitation for the hit events between the IMERG and RGs.

| No. | Site | IMERG GRID | GPCC GRID | Half-Hourly | | | Daily | | |
|-----|------|------------|-----------|------|------|------|------|------|------|
| | | | | NME | NMAE | CC | NME | NMAE | CC |
| 1 | Beibeng | IMERG01 | GPCC01 | −0.31 | 0.71 | 0.2 | −0.72 | 1 | −0.23 |
| 2 | Gelin | IMERG01 | GPCC01 | −0.28 | 0.79 | 0.1 | −0.42 | 1.31 | ***−0.2*** |
| 3 | Yarang | | GPCC01 | −0.22 | 0.72 | 0.16 | −0.45 | 1.16 | −0.18 |
| 4 | Dexing | | GPCC01 | −0.28 | 0.75 | 0.12 | −0.44 | 1.21 | ***−0.18*** |
| 5 | Wenlang | IMERG02 | GPCC01 | −0.25 | 0.73 | 0.18 | −0.23 | 1.36 | ***−0.18*** |
| 6 | Motog | IMERG02 | GPCC01 | −0.16 | 0.73 | 0.17 | −0.4 | 1.11 | ***−0.08*** |
| 7 | Bari | IMERG02 | GPCC01 | −0.35 | 0.72 | 0.24 | −0.12 | 1.49 | ***−0.24*** |
| 8 | Renqinbeng | IMERG02 | GPCC01 | −0.34 | 0.73 | 0.15 | −0.35 | 1.3 | ***−0.23*** |
| 9 | Miri | IMERG03 | GPCC01 | −0.31 | 0.69 | 0.11 | −0.27 | 1.23 | ***−0.12*** |
| 10 | Linduo | IMERG03 | GPCC01 | −0.47 | 0.71 | 0.16 | −0.8 | 0.9 | ***−0.12*** |
| 11 | Kabu | IMERG03 | GPCC01 | −0.33 | 0.68 | 0.14 | −0.4 | 1.19 | ***−0.13*** |
| 12 | Yimin | | GPCC01 | −0.09 | 0.75 | 0.14 | 0.05 | 1.48 | ***−0.18*** |
| 13 | Dongren | | GPCC01 | −0.16 | 0.75 | 0.15 | −0.44 | 1.15 | ***−0.06*** |
| 14 | 80K | | GPCC01 | −0.45 | 0.7 | 0.1 | −0.63 | 1.02 | ***−0.08*** |
| 15 | Xironggou | | GPCC01 | −0.16 | 0.87 | ***0.06*** | −0.55 | 1.05 | ***−0.07*** |
| 16 | Danka | | GPCC01 | 0.26 | 0.97 | 0.12 | −0.23 | 1.1 | ***−0.03*** |
| 17 | Bomi | | GPCC01 | —— | —— | —— | −0.34 | 1.16 | ***−0.13*** |
| 18 | Milin | | GPCC02 | —— | —— | —— | 0.11 | 1.36 | ***−0.08*** |
| 19 | Linzhi | | GPCC02 | —— | —— | —— | −0.44 | 1.02 | ***−0.04*** |
| 20 | Lulang | | GPCC02 | −0.06 | 0.82 | 0.11 | 0.64 | 2 | ***−0.09*** |
| 21 | Pailong | | GPCC03 | 0.34 | 0.89 | ***0.03*** | −0.12 | 1.1 | ***−0.09*** |
| 22 | Luolong | | GPCC03 | —— | —— | —— | 0.28 | 1.47 | ***0.01*** |

Notation: The sites are numbered from south to north. NME, normalized mean error; NMAE, normalized mean absolute error; CC, correlation coefficient. The bold and italicized numbers denote the sites that do not pass the significance test. "——" indicates no data.

IMERG exhibits a systematic negative bias (NME < 0) of the half-hourly rainfall (ranging from −0.06 to −0.45), thereby underestimating the rainfall in the YGC area (Table 4). This negative bias is larger at lower altitudes and decreases with increasing altitude, which is due to the fact that sites located at lower elevations are usually at the southern end of the range, where there is more water vapor.

When the rainfall is accumulated on a daily scale (Table 4), the systematic negative bias is compounded, leading to an increase in IMERG's negative bias at most stations. In addition, the random error on the daily scale is also amplified during the accumulation, leading to larger random errors (NAME 0.9 to 2) for all of the RGs on the daily rainfall scale. This makes it difficult for the correlation between IMERG and the RGs at the daily rainfall scale to pass the significance test. Hence, the negative correlation itself is not reliable. In conclusion, despite the successful prediction of rainfall events, IMERG's rainfall estimation still presents a significant systematic negative bias and substantial random errors. Compared to the daily rainfall scale, IMERG performs better on the half-hourly scale, providing more possibilities for short-term weather studies in the YGC area.

### 3.3. Influence of Terrain on IMERG Detection

As a typical orographically induced precipitation region, the YGC area exhibits a spatial precipitation gradient that changes with altitude (Figure 2). We selected the elevation and the relative elevation as the terrain factors to investigate the correlations between IMERG's evaluation indicators and the terrain (Figures 4–6). Considering the similarities of IMERG products subjected to monthly-scale correction under the same GPCC grid and the potential influence of RG data from different sources, we divided the data into three categories for the correlation computation: (1) r-all, representing all INVC-RGs and

CMA-RGs points; (2) r-st18, only INVC-RGs points; and (3) r-st16, INVC-RGs points within the GPCC01 grid.

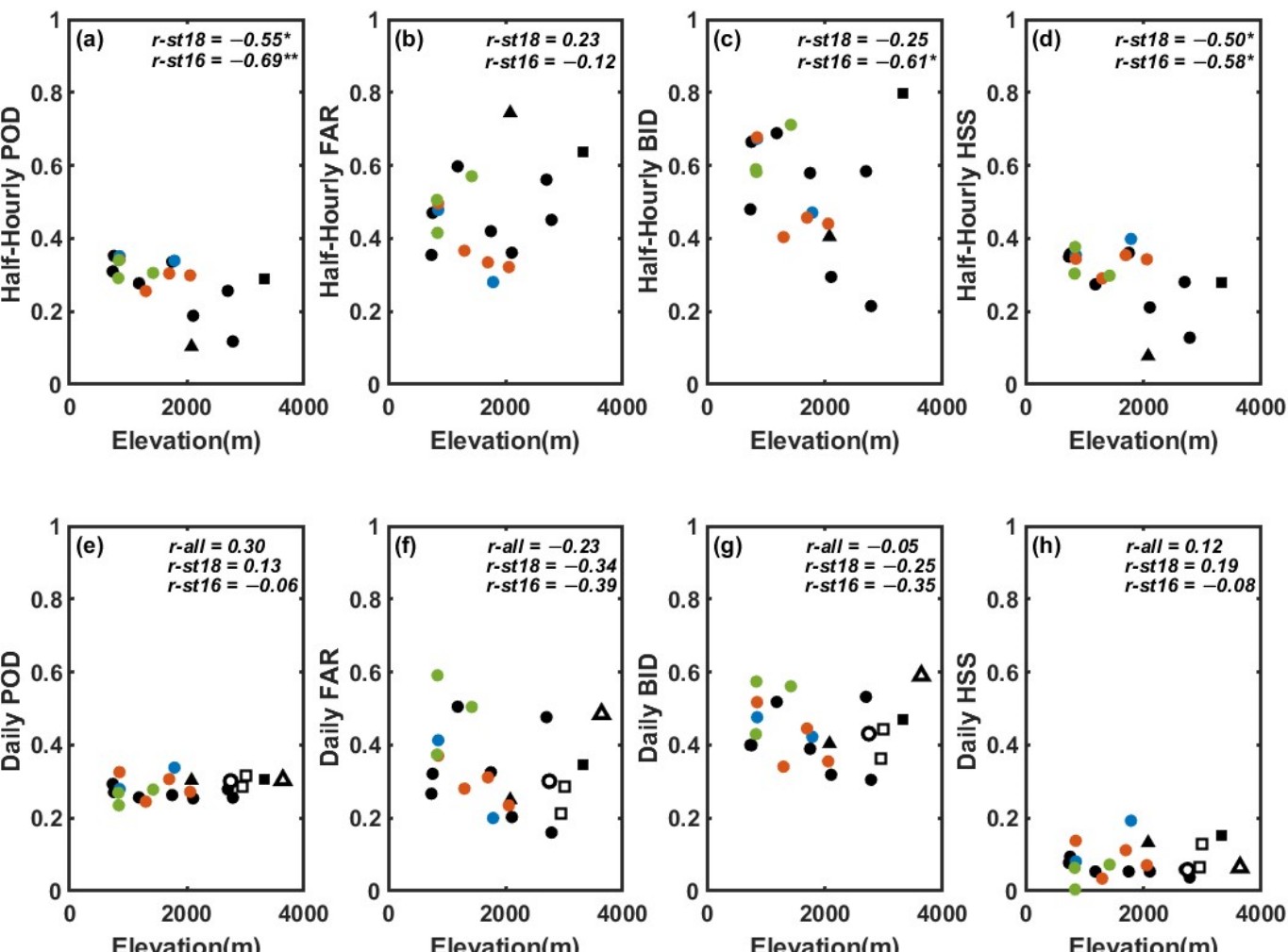

**Figure 4.** Scatter plots of the elevations (unit: m) of the sites versus the four scores (POD, FAR, BID, and HSS) at the (**a–d**) half-hourly and (**e–h**) daily scales. The circles represent the sites in the GPCC01 grid, the squares represent sites in the GPCC02 grid, and the triangles represent the sites in the GPCC03 grid. The hollow marks indicate the stations operated by the China Meteorological Agency (CMA), and the solid marks indicate non-CMA sites. The blue circles indicate the IMERG01 grid, the red circles indicate the IMERG02 grid, and the green circles indicate the IMERG03 grid. The correlations (r) between the four scores and the DEMs are noted in the upper right corner of each panel: r-all represents the correlations derived from all 22 sites at the daily scale, r-st18 represents the correlations derived from the 18 non-CMA sites, and r-st16 represents the correlations of the 16 non-CMA sites in the GPCC01 grid. * indicates a *p*-value of 0.05, and ** indicates a *p*-value of 0.01.

Figure 4 illustrates the relationship between IMERG's ability to capture precipitation events and the elevation. As the elevation increases, the ability of IMERG to detect precipitation events (POD) decreases, the underestimation of the precipitation events (BID) increases, and hence, the accuracy and reliability (HSS) of the satellite precipitation estimation decrease. The impact of elevation primarily occurs on the half-hourly scale, and as the temporal step accumulates, this correlation becomes less noticeable. The relationship between the FAR and elevation is somewhat unclear, which may be associated with the limited number of stations. Future work may investigate this issue by increasing the number of high-altitude sites.

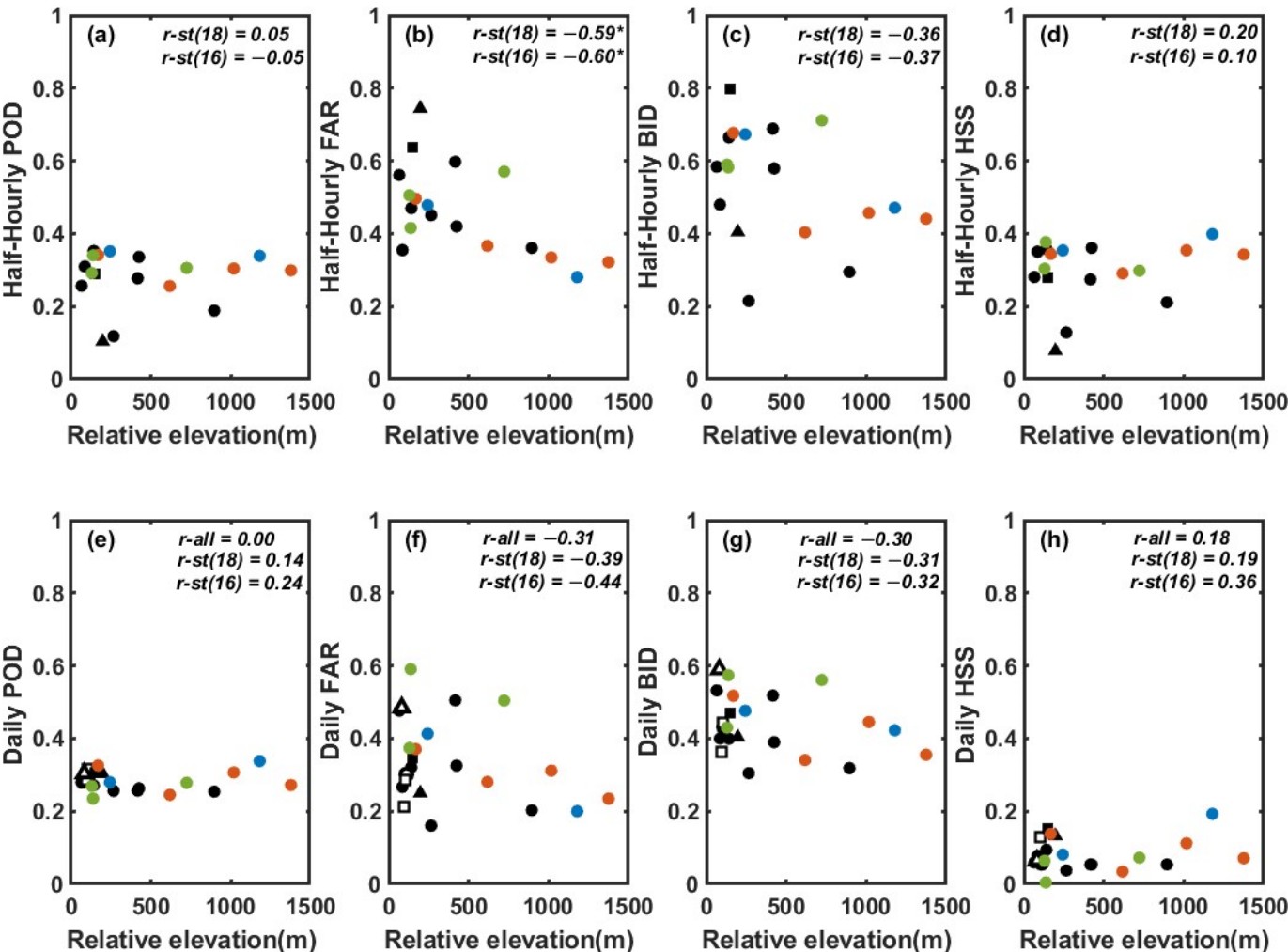

**Figure 5.** Scatter plots of the relative elevations (unit: m) of the sites versus the four scores (POD, FAR, BID, and HSS) at the (**a–d**) half-hourly and (**e–h**) daily scales. The circles represent the sites in the GPCC01 grid, the squares represent sites in the GPCC02 grid, and the triangles represent the sites in the GPCC03 grid. The hollow marks indicate the stations operated by the China Meteorological Agency (CMA), and the solid marks indicate non-CMA sites. The blue circles indicate the IMERG01 grid, the red circles indicate the IMERG02 grid, and the green circles indicate the IMERG03 grid. The correlations (r) between the four scores and the DEMs are noted in the upper right corner of each panel: r-all represents the correlations derived from all 22 sites at the daily scale, r-st18 represents the correlations derived from the 18 non-CMA sites, and r-st16 represents the correlations of the 16 non-CMA sites in the GPCC01 grid. * indicates a *p*-value of 0.05.

As shown in Figure 5b, we found a significant negative correlation (−60%) between the FAR and relative elevation in the case of half-hourly precipitation events, implying that IMERG has a higher FAR in the valley sites. By dividing the half-hourly data into day and night, we could see the difference in the correlation between day and night (Figure A1). The negative correlation between FAR and relative elevation is more pronounced during the daytime. It might be due to greater mountain–valley wind speeds amplifying the effect of topography on FAR during the day.

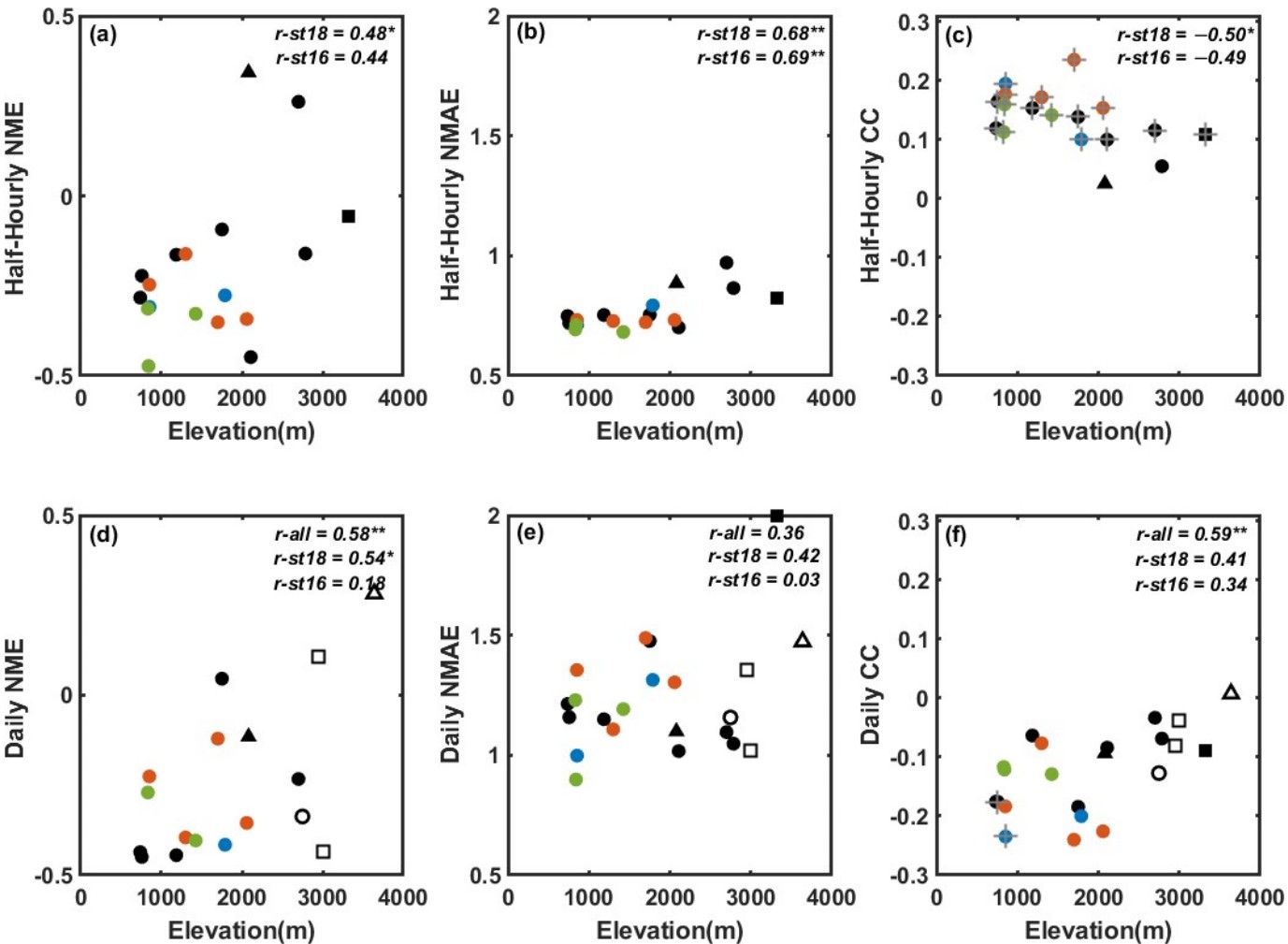

**Figure 6.** Scatter plots of the elevations (unit: m) of the sites versus the normalized mean error (NME), normalized mean absolute error (NMAE), and correlation coefficient (CC) at the (**a**–**c**) half-hourly and (**d**–**f**) daily scales. The circles represent the sites in the GPCC01 grid, the squares represent sites in the GPCC02 grid, and the triangles represent the sites in the GPCC03 grid. The hollow marks indicate the stations operated by the China Meteorological Agency (CMA), and the solid marks indicate non-CMA sites. The blue circles indicate the IMERG01 grid, the red circles indicate the IMERG02 grid, and the green circles indicate the IMERG03 grid. The correlations (r) between the four scores and the DEMs are noted in the upper right corner of each panel: r-all represents the correlations derived from all 22 sites at the daily scale, r-st18 rep-resents the correlations derived from the 18 non-CMA sites, and r-st16 represents the correlations of the 16 non-CMA sites in the GPCC01 grid. * indicates a *p*-value of 0.05, and ** indicates a *p*-value of 0.01. The grey + symbol on the circles in (**c**,**f**) indicates that the site reaches a *p*-value of 0.05.

Next, we focused on hit events and analyzed the NME, NMAE, and CC with respect to the site's elevation and relative elevation. While elevation shows some correlations with these metrics, the relative elevation demonstrates no more significant correlations, so we only show the relationship between the metrics and elevation in Figure 6. These results indicate that the influence of the site's position on whether IMERG identifies precipitation events as occurring outweighs its influence on the estimated rainfall volume. As can be seen from Figure 6a,d, even if IMERG correctly captures the occurrence of precipitation events, there is still a systematic negative bias at lower altitude sites in terms of rainfall estimation. As the altitude increases, this negative bias gradually decreases. Within the 2000–4000 m altitude range, IMERG's systematic bias is close to 0 at some stations

while exhibiting a significant positive bias for others. The half-hourly NMAE exhibits a strong positive correlation with the elevation (up to 68%) (Figure 6b), implying that IMERG has a larger random error in high-altitude areas. A larger random bias occurred in daily precipitation than that on half-hourly scales (Figure 6e). Correspondingly, IMERG's NMAE is correlated with the CC for the half-hourly precipitation, but there is not a reliable correlation between the CC and DEM for the daily rainfall. Thus, the reliability of IMERG's half-hourly precipitation is higher.

Overall, we found that the ability of IMERG to detect precipitation events decreases with increasing elevation, and IMERG exhibits a higher FAR in valley areas. In hit events, increasing elevation improves the negative bias of IMERG rainfall.

### 3.4. IMERG Performance under Different Rainfall Intensities

We examined IMERG's ability to capture various rainfall intensities and the distribution of the accumulated precipitation when precipitation events are hit by IMERG (Figure 7). Figure 7a demonstrates that IMERG tends to overestimate the occurrence of light and moderate rainfall events while underestimating the occurrence of events with rainfall intensities exceeding 0.8 mm/0.5 h. This pattern is also reflected in the accumulated rainfall estimates (Figure 7b). IMERG performs commendably for extreme rainfall ($p > 3.2$ mm/0.5 h) in terms of both the number of precipitation events and the comparison of the accumulated rainfall. Overall, IMERG tends to overestimate light to moderate rainfall events and underestimate heavy rainfall to rainstorm events, causing a shift in the precipitation distribution towards lower intensities and an overall underestimation bias. This phenomenon has also been observed in IMERG evaluations along the eastern coast of the U.S. and has been identified as a systematic bias in IMERG [5].

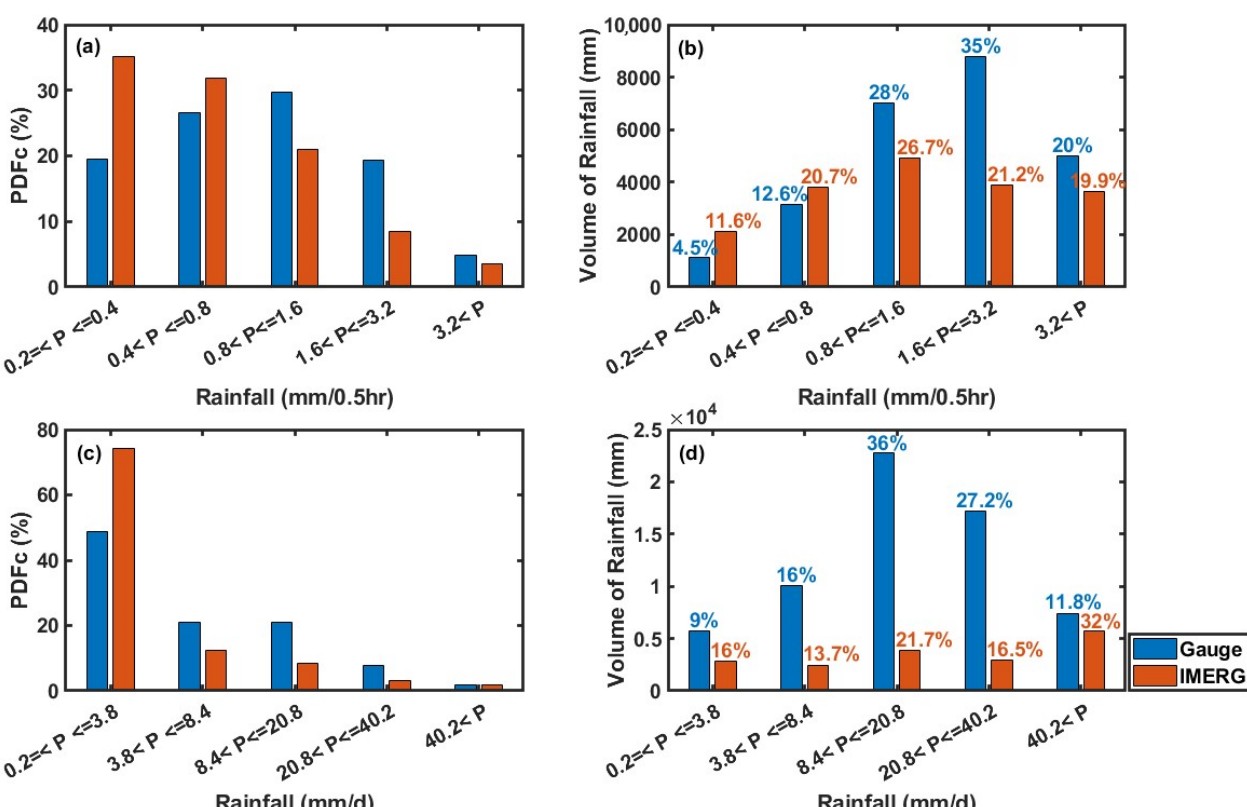

**Figure 7.** (**a**) Probability density function by occurrence (PDFc) and (**b**) Volume of rainfall and probability density function by volume (PDFv, the percentage number is on the histogram) of the half-hourly rainfall with different intensities for hit events. (**c**) PDFc and (**d**) Volume of rainfall and PDFv of daily rainfall. The blue bars represent the rain gauges, and the red bars represent IMERG.

After the precipitation is accumulated into the daily precipitation, the hit ratio increases, and the distribution of IMERG's daily rainfall shifts towards lighter precipitation (Figure 7c). IMERG overestimates the occurrence of light rainfall ($p \leq 3.8$ mm/d), and it underestimates moderate to heavy and extreme rainfall. It was found that IMERG underestimates the total rainfall amounts for all precipitation intensities (Figure 7d), and the distribution patterns of the accumulated precipitation of IMERG under different rainfall intensities exhibit a certain discrepancy compared to ground observations. For ground-based daily precipitation, heavy rainfall ($8.4 < p \leq 20.8$ mm/d) contributes the most (about 36%), followed by rainstorms, moderate rainfall, extreme rainfall, and light rainfall. However, the highest contributions in IMERG come from extreme rainfall, heavy rainfall, rainstorms, light rainfall, and moderate rainfall. Therefore, caution is advised when utilizing IMERG's daily precipitation data to study local rainfall intensity distributions.

*3.5. Diurnal Variation Assessment*

Prior studies, including those conducted by Zhou, Kim, and Guan [38], have acknowledged IMERG's effectiveness in representing large-scale diurnal rainfall patterns. They noticed that IMERG's diurnal and semidiurnal data mostly aligned with the reference values, but they also highlighted IMERG's tendency to underestimate the peak precipitation in mountainous regions and its propensity to detect precipitation prematurely in mesoscale convective systems. Chen et al. [2] revealed a distinct diurnal variation pattern within the YGC region, which differed from that on the central plateau. This pattern shows that the DVP reaches its lowest point at around 15:00 local standard time, then slowly increases and peaks in the early morning. Therefore, it is essential to examine whether IMERG can accurately capture this unique DVP in the YGC region.

The evaluation conducted in this paper is concentrated on two areas (Table 5): (1) the DVP phase, including the ability to capture the peak rainfall times and the level of synchronicity between the two datasets (indicated by the CC), and (2) the systematic bias of the DVP (indicated by the NME). The diurnal variation in Figure 8 was obtained by averaging the daily variation of all available data, and the CC and the NME were calculated based on the mean diurnal variation. IMERG is a comprehensive dataset that was produced by combining microwave sensor data, infrared data, and gauge calibration data. These individual datasets contribute positively to the final IMERG product, but they may also introduce their own biases. Hence, when analyzing the DVP, it is vital to assess not only the calibrated IMERG data (hereinafter referred to as Cal) but also the HQ, IR, and Uncal data.

By studying the NMEs of the HQ, IR, Uncal, and Cal datasets, we found that all of these datasets demonstrated negative biases, with only a few exceptions (such as Cal NME at Lulang station in Table 5). This implies a systemic underestimation of the rainfall at the sensor level. By comparing the HQ and IR data, we found that the IR generally exhibited a better correlation with the DVP (Table 5). IR data are constantly gathered, providing regular updates on the presence or absence of rainfall at specific time intervals. In contrast, HQ data are collected via sporadic microwave detection. As a result, HQ data often exhibit poorer correlation coefficients and, in some cases, even uncorrelated results (bold and italicized in Table 5), particularly at stations experiencing pronounced diurnal variations and larger Cal negative biases. When the HQ and IR data were combined to generate the Uncal data, we noticed an improvement in the negative bias and an increase in the CC. This suggests that the integration of microwave and infrared data improves the chances of detecting rainfall events, thereby enhancing the depiction of the diurnal phase. By comparing the Cal and Uncal data, it was found that there was no significant change in the CC, but the negative bias was substantially improved. For instance, at Lulang station, it increased from −32% to 5%. This confirms that the GPCC's primary aim is to fine-tune the rainfall volume rather than altering its occurrence.

**Table 5.** Evaluation of IMERG diurnal variation of the precipitation (DVP).

| No. | Site | IMERG GRID | GPCC GRID | HQ | | IR | | Uncal | | Cal | |
|-----|------|-----------|-----------|------|------|------|------|------|------|------|------|
| | | | | NME | CC | NME | CC | NME | CC | NME | CC |
| 1 | Beibeng | IMERG01 | GPCC01 | −0.7 | 0.64 | −0.69 | 0.43 | −0.66 | 0.74 | −0.41 | 0.7 |
| 2 | Gelin | IMERG01 | GPCC01 | −0.75 | 0.59 | −0.75 | 0.41 | −0.72 | 0.73 | −0.51 | 0.69 |
| 3 | Yarang | | GPCC01 | −0.66 | 0.56 | −0.66 | 0.59 | −0.6 | 0.79 | −0.33 | 0.75 |
| 4 | Dexing | | GPCC01 | −0.71 | 0.51 | −0.71 | *0.39* | −0.71 | 0.71 | −0.52 | 0.73 |
| 5 | Wenlang | IMERG02 | GPCC01 | −0.61 | 0.58 | −0.64 | 0.53 | −0.65 | 0.85 | −0.42 | 0.87 |
| 6 | Motog | IMERG02 | GPCC01 | −0.65 | 0.51 | −0.68 | 0.66 | −0.69 | 0.77 | −0.48 | 0.76 |
| 7 | Bari | IMERG02 | GPCC01 | −0.68 | 0.67 | −0.71 | 0.47 | −0.72 | 0.9 | −0.53 | 0.92 |
| 8 | Renqinbeng | IMERG02 | GPCC01 | −0.73 | 0.68 | −0.75 | 0.69 | −0.76 | 0.91 | −0.6 | 0.9 |
| 9 | Miri | IMERG03 | GPCC01 | −0.56 | 0.55 | −0.64 | 0.62 | −0.58 | 0.78 | −0.36 | 0.74 |
| 10 | Linduo | IMERG03 | GPCC01 | −0.66 | 0.63 | −0.72 | 0.6 | −0.68 | 0.83 | −0.51 | 0.81 |
| 11 | Kabu | IMERG03 | GPCC01 | −0.48 | 0.58 | −0.58 | 0.64 | −0.51 | 0.78 | −0.25 | 0.74 |
| 12 | Yimin | | GPCC01 | −0.54 | *0.32* | −0.65 | 0.74 | −0.55 | 0.82 | −0.34 | 0.79 |
| 13 | Dongren | | GPCC01 | −0.43 | *0.24* | −0.5 | 0.43 | −0.43 | 0.5 | −0.16 | 0.49 |
| 14 | 80K | | GPCC01 | −0.85 | 0.5 | −0.85 | 0.55 | −0.83 | 0.77 | −0.75 | 0.78 |
| 15 | Xironggou | | GPCC01 | −0.86 | *0.4* | −0.85 | 0.55 | −0.84 | 0.84 | −0.76 | 0.83 |
| 16 | Danka | | GPCC01 | −0.51 | *0.39* | −0.49 | *0.26* | −0.48 | 0.6 | −0.17 | 0.56 |
| 20 | Lulang | | GPCC02 | −0.4 | *0.11* | −0.35 | 0.6 | −0.32 | 0.56 | 0.05 | 0.53 |
| 21 | Pailong | | GPCC03 | −0.56 | *0.09* | −0.57 | *0.19* | −0.56 | *0.37* | −0.3 | 0.44 |

Notation: HQ, GPM microwave; IR, GPM infrared; Uncal, Un-calibrated IMERG; Cal, IMERG calibrated; NME, normalized mean error; CC, correlation coefficient. The bold and italicized numbers denote that the sites do not pass the significance test. The sites are numbered from south to north.

Figure 8 illustrates that both IMERG and the RGs commonly display an early morning surge in rainfall. This can be attributed to a mix of high-frequency and high-intensity rainfall [2]. The DVP at the RGs is primarily manifested in two forms. (1) Concentrated precipitation peak: The minimum rainfall tends to occur around 16:00, and then, the rainfall escalates and reaches a peak between 3:00 and 7:00. This pattern is observable at Site No.1−3, 9−14, 16, and 21 (Figure 8a,b,e–j,l). (2) Multiple precipitation peaks: The primary peak still occurs in the early morning, but it is accompanied by various secondary peaks (as observed in Site No.4−8, 15, and 20 in Figure 8c,d,i,k). IMERG tends to represent multiple peaks. As the stations are positioned along the trajectory of the water vapor transmission, the time of the rainfall peak is delayed from south to north, with IMERG predominantly centering the rainfall peak at around 4:00. Thus, the satellite data often struggle to capture the variations in the precipitation peaks over time.

We observe that the stations with higher correlation coefficients (i.e., CC > 0.6) typically experienced larger rainfall volumes. This implies that increased rainfall at a station simplifies the task for IMERG in identifying DVP patterns. In contrast, the stations with less daily rainfall and a flatter DVP tended to have lower CCs. In the IMERG02 grid, four stations share the same basin and water vapor channel and are devoid of disturbances from other water vapor sources. As the altitude increases, the rainfall measured by the RG increases, transitioning from the valley to the slope. This increases the negative bias and enhances IMERG's correlation with the DVP. Thus, within the same IMERG grid, satellites can more readily capture the DVP volume at valley stations. However, their proficiency in reflecting the phase of the diurnal variations is not as good for stations on slopes or peaks. This is consistent with our prior discussion of the satellite's capability to pinpoint the timing of a rainfall event.

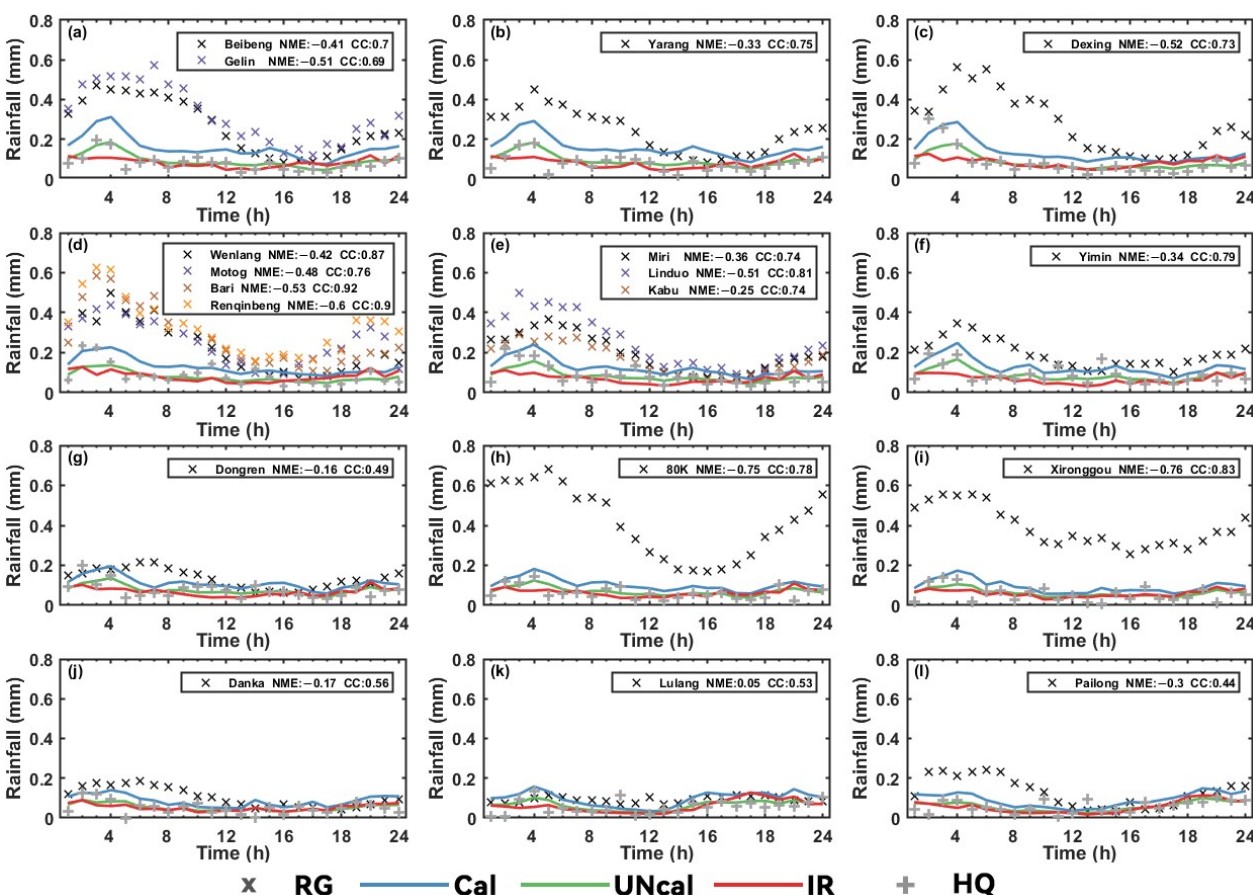

**Figure 8.** The diurnal variation in the rainfall derived from the rain gauges and IMERG. Each panel shows the result of one IMERG grid. The x symbols represent the rain gauge data, the blue lines represent the Cal data (IMERG calibrated), the green lines represent the Uncal data (Un-calibrated IMERG), the red lines represent the IR data (GPM infrared rainfall), and the grey + symbols represent the HQ data (GPM microwave rainfall). The Correlation coefficient (CC) and normalized mean error (NME) of the Cal data against the rain gauge data are noted in the upper right corner of each panel. All of the times are Beijing time. (**a**,**d**,**e**) correspond to IMERG01, IMERG02, IMERG03, respectively. (**a**–**g**) correspond to stations south of Dongren, and (**h**–**l**) correspond to stations north of 80K.

### 3.6. Assessment of Seasonal Variation

We have confirmed that the calibration process using GPCC data enhanced the accuracy of IMERG outputs relative to station observations. Regarding the calibration process, it is evident that the GPCC's monthly data significantly influences the quality of the IMERG data across different time scales. This relationship is validated in Table 6; that is, the correlation between the GPCC and IMERG exceeds 92%, and the relative bias ranges from −29% to +32%. As a result, any bias between the GPCC and station measurements would directly affect the quality of the IMERG data.

Table 6 shows that there is considerable variability in the NME and CC between the GPCC and the various stations, which suggests a certain level of representational issue with the GPCC's 1° resolution at the station level. Therefore, it is crucial to offer a higher-resolution GPCC product for IMERG within the YGC region. The seasonal traits depicted by the GPCC (Figure 9) show that during the monsoon season (June–September), the main rainfall peak is in July, and there is a secondary peak in September. In the pre-monsoon period (March–May), a weak peak occurs in April. These seasonal features are also reflected in IMERG's seasonal variations (Figure 10). The seasonal variation at some RG stations corresponds well with IMERG's representation, with the CC reaching 0.99 at Linzhi station (Table 6). However, discrepancies between the seasonal variation at some stations and

satellite estimates do exist, such as during the monsoon season with the primary peak in June (Renqinbeng, Yimin in Figure 10d,f), the primary peak in September (Pailong in Figure 10o), and pre-monsoon peaks in April (Dexing, Wenlang, Renqinbeng, Yimin, and Danka in Figure 10c,d,f,j) and March (Dongren in Figure 10g).

**Table 6.** Evaluation of seasonal variations of the GPCC against IMERG, the GPCC against gauge stations, and IMERG against gauge stations.

| No. | Site | IMERG GRID | GPCC GRID | GPCC vs. IMERG | | GPCC vs. Stations | | IMERG vs. Stations | |
|---|---|---|---|---|---|---|---|---|---|
| | | | | NME | CC | NME | CC | NME | CC |
| 1 | Beibeng | IMERG01 | GPCC01 | −0.28 | 0.96 | −0.67 | 0.95 | −0.54 | 0.95 |
| 2 | Gelin | IMERG01 | GPCC01 | −0.28 | 0.96 | −0.63 | 0.75 | −0.49 | 0.84 |
| 3 | Yarang | | GPCC01 | −0.29 | 0.95 | −0.62 | 0.93 | −0.47 | 0.96 |
| 4 | Dexing | | GPCC01 | −0.16 | 0.94 | −0.59 | 0.88 | −0.52 | 0.87 |
| 5 | Wenlang | IMERG02 | GPCC01 | −0.15 | 0.95 | −0.52 | 0.88 | −0.44 | 0.97 |
| 6 | Motog | IMERG02 | GPCC01 | −0.15 | 0.95 | −0.72 | 0.96 | −0.67 | 0.93 |
| 7 | Bari | IMERG02 | GPCC01 | −0.15 | 0.95 | −0.59 | 0.8 | −0.52 | 0.92 |
| 8 | Renqinbeng | IMERG02 | GPCC01 | −0.15 | 0.95 | −0.66 | 0.87 | −0.6 | 0.89 |
| 9 | Miri | IMERG03 | GPCC01 | −0.08 | 0.95 | −0.57 | 0.97 | −0.53 | 0.96 |
| 10 | Linduo | IMERG03 | GPCC01 | −0.08 | 0.95 | −0.6 | 0.78 | −0.57 | 0.77 |
| 11 | Kabu | IMERG03 | GPCC01 | −0.08 | 0.95 | −0.56 | 0.94 | −0.52 | 0.97 |
| 12 | Yimin | | GPCC01 | −0.11 | 0.94 | −0.39 | 0.82 | −0.32 | 0.88 |
| 13 | Dongren | | GPCC01 | 0.02 | 0.96 | −0.42 | 0.76 | −0.43 | 0.79 |
| 14 | 80K | | GPCC01 | 0.1 | 0.95 | −0.76 | 0.95 | −0.78 | 0.94 |
| 15 | Xironggou | | GPCC01 | 0.18 | 0.92 | −0.74 | 0.97 | −0.78 | 0.94 |
| 16 | Danka | | GPCC01 | 0.29 | 0.94 | 0.09 | 0.89 | −0.16 | 0.89 |
| 17 | Bomi | | GPCC01 | 0.32 | 0.95 | 0.01 | 0.89 | −0.23 | 0.86 |
| 18 | Milin | | GPCC02 | −0.08 | 0.99 | 0.12 | 0.96 | 0.22 | 0.96 |
| 19 | Linzhi | | GPCC02 | 0.15 | 0.99 | 0.1 | 0.99 | −0.04 | 0.99 |
| 20 | Lulang | | GPCC02 | 0.08 | 0.98 | −0.33 | 0.96 | −0.38 | 0.96 |
| 21 | Pailong | | GPCC03 | −0.07 | 0.97 | −0.5 | 0.95 | −0.47 | 0.9 |
| 22 | Luolong | | GPCC03 | 0.25 | 0.95 | 0.31 | 0.94 | 0.05 | 0.98 |

Notation: NME, normalized mean error; CC, correlation coefficient. The sites are numbered from south to north.

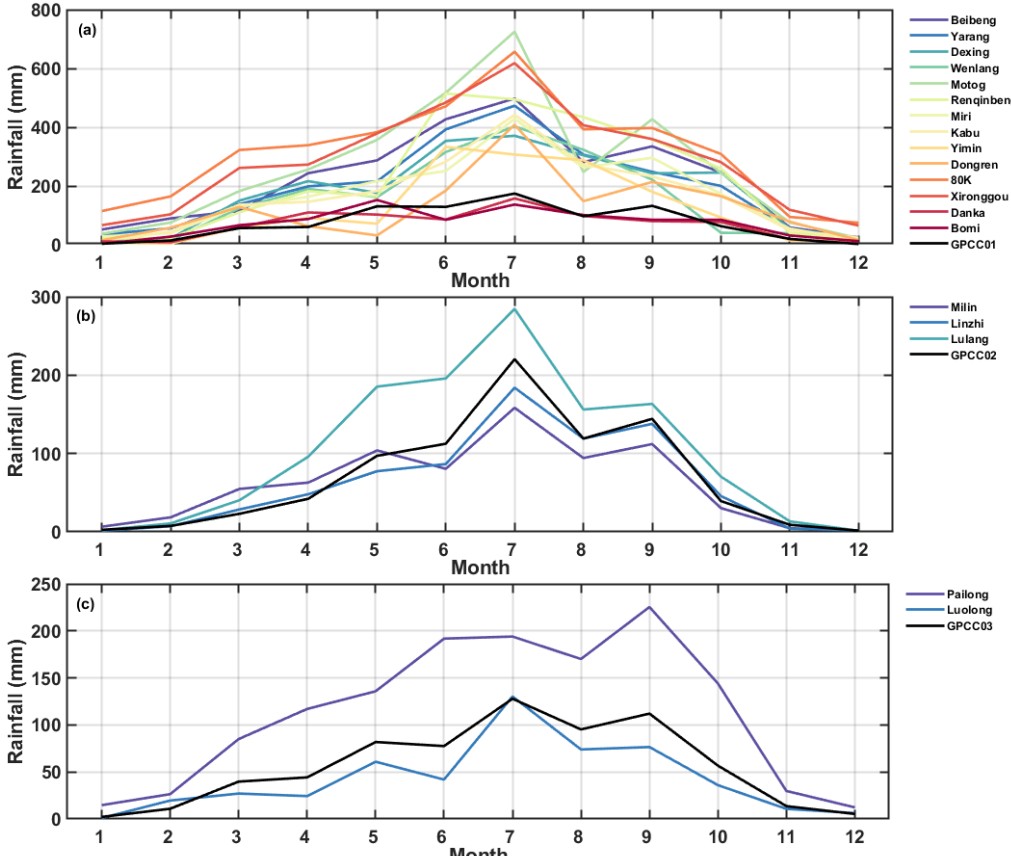

**Figure 9.** Seasonal variation comparison of rain gauges and GPCC grid value for (**a**) grid GPCC01, (**b**) grid GPCC02, and (**c**) grid GPCC03.

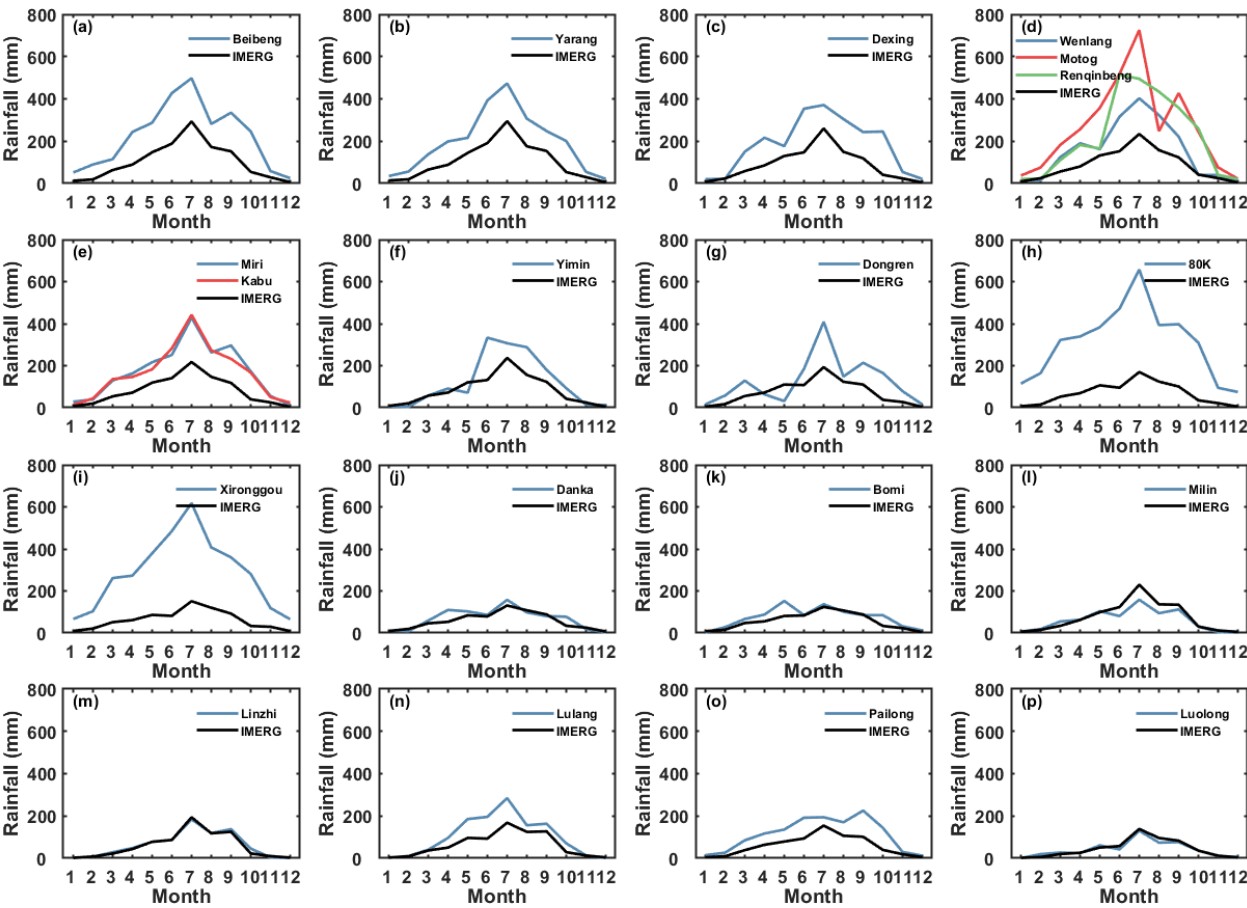

**Figure 10.** Seasonal variation comparison of rain gauges and IMERG grid value. Each panel shows the result of one IMERG grid. The sites are numbered from south to north. (**a**–**p**) correspond to the different IMERG grids.

## 4. Discussion

An evaluation of IMERG's performance in relation to rainfall events, quantity of rainfall, rainfall intensity, diurnal variations, and seasonal variations reveals that its applicability in the YGC region is influenced by a combination of the region's hydrometeorological characteristics and topography. The results are discussed in terms.

(1)    The underestimation in IMERG exists at the GPM sensor level.

In the IMERG observational system, the quality of IMERG data is influenced by the PMW sensor, the IR data, and the gauge data used for calibration. Our evaluation demonstrated that the ratio of the miss rate is higher than the false alarm, which is also detected on the U.S. Eastern Shore [5]. Tan et al. [5] pointed out that this is associated with the PMW instruments. The performance of the onboard Ku-PR is essential for ensuring data quality. Yet, significant under-detection errors were noted in Ku-PR evaluations [37]. Arulraj and Barros [37] pinpointed two main factors causing this underestimation: firstly, ground clutter impacts on the 1–2 km reflectivity profile, lead to non-uniform beam filling, causing the underestimation of low-level and bottom-level precipitation systems. Secondly, seeder-feeder interactions are prevalent in mountainous regions when two layers of cloud happen. Zhou et al. [39] reported a bimodal distribution of cloud base heights in the YGC region. The seeder-feeder interactions might have a high chance of happening in this region, which generally leads to enhanced precipitation recorded by rain RGs near minor mountain areas [40,41]. Ku-PR struggles to capture these enhanced lower-level echoes from the seeder cloud [37]. This explains why the GPM has an underestimation for this region. Wang et al. [42] reported that the raindrop size distribution in Motog was

maritime-like. However, the raindrop size distribution in the GPM-IMERG algorithm for the YGC region might be a different one. Uncertainties in raindrop size distribution can lead to satellite precipitation underestimations [37]. As such, future research should intensify observations on cloud microphysics characteristics within the YGC region to formulate a specific raindrop spectral parameter tailored to its conditions.

IMERG's tendency to overestimate light rainfall events and underestimate heavy rainfall events can be traced back to issues at the instrument level, observed in PMW sensors such as TMI, the Special Sensor Microwave Imager/Sounder (SSMIS), and the Microwave Humidity Sounder (MHM) [5]. Additionally, the integration of PMW with geo-IR estimates happens in two ways [14]: First, morphing the LEO-PMW data. Second, in cases where LEO-PMW data are sparse, geo-IR precipitation estimates are incorporated using a Kalman filter. Tan et al. [5] observed that part of the IR's underestimation can be attributed to the morphing process, which tends to propagate the PMW's underestimation to the IR data. Both underestimation of the PMW and IR data was observed by us in the diurnal variations (as seen in Figure 8); hence, the Uncal, which is the combination of PMW (HQ) and IR, also revealed an underestimation.

(2) Higher FAR in the valley stations.

The negative correlation between FAR and the relative elevation was observed in Figure 5, which represents the FAR in the valley is higher than that at the mountain peaks. The water vapor in the valley largely stems from warm, moist air masses transported from the Indian Ocean [42]. When these moist air masses from the Indian Ocean enter into the valley, they might not always reach the lifting condensation level to generate precipitation immediately and form stratiform clouds or fog in the valley. Wang et al. [42] demonstrated that stratiform rainfall constitutes over 95% of total events at Motog station. Arulraj and Barros [37] reported that the GPM satellite has high false alarms for stratiform rainfall. Due to orographic lifting—the process where terrain forces the air to rise—these moist air masses are channeled up the slopes, inducing more precipitation at higher elevations. This orographic effect augments the probability of IMERG misidentifying non-precipitating clouds as rain clouds in the valley.

In scenarios of light precipitation, the spatiotemporal intermittency is prominent. If fewer PMWs traverse the valley region, IMERG might lean towards using the instantaneous precipitation rate, deduced linearly from PMW signals bracketing the 30 min interval. This method might occasionally lead to satellite estimates indicating rain, even when ground observations at the valley affirm a lack of precipitation. Further corroborating this, Arulraj and Barros [37] pinpointed the morphing of PMW data as a chief contributor to false alarms, culminating in misguided rainfall indications during clear periods. These factors can be invoked to elucidate the pronounced negative correlation between FAR and relative elevation.

(3) In the evaluation of rainfall events, why do topographic correlations stand out on half-hourly scales but fade on daily scales?

The POD, FAR, BID, and HSS have a significant correlation with terrains on half-hourly scales, but that is not significant on daily scales. Statistically, the POD represents IMERG's capability to detect precipitation events within a set timeframe. The odds of the satellite capturing rainfall events rise with the increasing time scales. For instance, if the half-hourly POD of site A and site B is 50% and 30%, the daily POD of the two sites will be 100%. This indicates that the variability of POD, influenced by elevation, becomes negligible on a daily scale. The YGC region, serving as a crucial conduit for water vapor transport, is molded by both expansive weather systems such as the South Asian monsoon and westerlies, as well as terrain-induced circulations of various scales. On the half-hour scale, terrain-induced circulations result in noticeable spatial heterogeneity. However, the spatial pattern of precipitation on the daily scale is more homogeneous. This means that the variations of POD, FAR, BID, and HSS among different sites could be diminished from the half-hourly to daily scale.

(4)     The benefit of GPCC gauge calibration in IMERG algorithm.

The GPCC exhibits a substantial negative bias south of Galongla Mountain (sites No. 1–15). However, in the northern region of the mountain, the bias is relatively minor. This is mainly due to the fact that the southern part of Galongla Mountain has long suffered from a dearth of rain gauge observation data before the establishment of the INVC-RGs. Conversely, the presence of CMA-RGs in the northern part of Galongla Mountain has enhanced the quality of GPCC and further improved IMERG-Final products in these areas. Thus, satellite validation work in the downstream region of the YGC is imperative and urgently needed.

By comparing the INVC-RG and CMA-RG in GPCC02 and GPCC03 grids, we identified the importance of choosing precipitation observation stations (Figure 9b,c). Lulang and Linzhi are located in the same grid of GPCC02 and influenced by the same water vapor branch originating from the YGC valley, but the value of the GPCC02 grid is very close to Linzhi but differs from that at Lulang (Table 6). This is primarily because Lulang station, situated on the northwestern side of the Sejila mountain, is affected by the orographic precipitation enhancement, leading to greater rainfall, while Linzhi, located on the leeward side of the Sejila mountain, experiences a decrease in the water vapor content, resulting in less rainfall than that at Lulang (Figure 1b). These geographic characteristics cause a different performance of IMERGs for the two stations. Pailong and Luolong are located in the same grid of GPCC03; the GPCC03 grid values closely match Luolong's but differ significantly from Pailong's. The GPM-IMERG grid value is closer to the CMA-RG sites (Linzhi and Luolong) than our INVC-RG sites (Lulang and Pailong); this demonstrates that the CMA-RG site has benefited from GPCC and GPM-IMERG precipitation estimation. YGC canyons are deep and broad, and thus, the precipitation formation mechanism is complex. The establishment of stations such as INVC-RGs is of significant importance as they enrich the spatial patterns of the precipitation under complex terrains and enhance the quality of the GPCC and IMERG in the YGC region.

A high correlation between GPCC and IMERG in Table 6 indicates that the 1° adjustment causes the seasonal variations of IMERG within the same GPCC grid to homogenize. Corrections should be considered starting from the data sources, such as the sensors. Moreover, when different platform estimates of the GPM are calibrated with each other and when Morph compensates for the absence of PMW, new errors are introduced. If these errors can be controlled, the effectiveness of the calibration using the GPCC can be enhanced.

Moving forward, our endeavors will pivot around the following key directions: First, Enhancing Representativeness of rain gauge data in the YGC Region. Obtaining ground-based data in the YGC region has always posed challenges. Our current dataset, though valuable, is relatively short in terms of temporal coverage, potentially limiting its climatic representativeness. We aim to gather data spanning longer periods in our subsequent endeavors, with the hope of achieving a more representative evaluation of IMERG. Second, we are exploring a range of methods to refine the calibration of IMERG data. For instance, we're currently testing a bias removal method, cumulative distribution functions (CDF) matching, to enhance the IMERG data's accuracy. At last, we are geared towards observing precipitation microphysics characteristics to fine-tune parameters in the GPM-IMERG algorithm. By focusing on these aspects, we aim to further the accuracy of GPM-IMERG data in the YGC region.

## 5. Conclusions

Conventional precipitation observations face challenges in accurately estimating precipitation in complex terrain and at high temporal resolutions. Satellite-based precipitation retrievals, such as IMERG, provide an alternative data source for filling these gaps. In this study, IMERG improved accuracy and detection capabilities compared to its predecessor, TRMM. However, IMERG still exhibited limitations in certain regions and for some precipitation event types. In this study, the performance of the IMERG half-hourly and daily precipitation products in the YGC region was evaluated. Furthermore, ground-based

precipitation data from observation stations, including the STEP-INVC program and the China Meteorological Administration, were used for validation. The evaluation results were as follows.

(1) The assessment of precipitation events in the YGC region revealed the limitations of IMERG in accurately detecting and estimating rainfall. The evaluation showed that IMERG exhibited errors in rainfall detection, including misses and false alarms, with misses being more prevalent than false alarms. This under-detection issue was found to persist across all sites in the YGC region. The comprehensive skill score of IMERG for daily rainfall events was less than ideal. Elevation affected IMERG's ability to detect precipitation events, with higher elevations leading to a decreased POD and increased underestimation. Valley areas exhibited lower POD, higher FAR, and lower HSS values, indicating less reliable precipitation event capture by IMERG in valleys. The station's position in a valley or on a slope influenced IMERG's performance, primarily impacting the detection of precipitation events rather than the estimated rainfall volume.

(2) We focused on rainfall events correctly hit by IMERG and assessed the rainfall amounts at the half-hour and daily scales. IMERG tended to underestimate rainfall amounts, particularly at lower altitudes. The negative bias decreased with increasing altitude and was influenced by the position of the station within the moisture transport pathways. Gauges located in tributaries had larger random errors and higher false alarm rates, contributing to the overall estimation errors of IMERG. Stations with dramatic changes in altitude gradient had high precipitation levels, leading to a lower correlation with IMERG. When rainfall accumulated on a daily scale, the negative bias was amplified, and the random errors increased. IMERG also tended to overestimate the occurrence of light and moderate rainfall events and underestimate events with higher intensities. This pattern was also reflected in the accumulated rainfall estimates.

(3) IMERG could capture the peak precipitation in the early morning diurnal variation in the YGC region but underestimated the amplitude of the diurnal variation. The datasets from various data sources (HQ, IR, UN, and Cal) exhibited negative biases, indicating a systemic underestimation of the rainfall at the sensor level. Combining HQ and IR data (Uncal) improved the negative bias and correlation and enhanced the depiction of the diurnal phase. The calibrated data (Cal) primarily adjusted the volume of the rainfall rather than its occurrence. IMERG was more proficient in capturing the DVP volume at stations in valleys than those on slopes or peaks.

(4) IMERG and GPCC could describe the seasonal characteristics at gauge sites while underestimating the seasonal variation at the sites. There was considerable variability in the NME and CC between the GPCC and various RGs, indicating representational issues with the GPCC's 1° resolution at the station level. Therefore, a higher-resolution GPCC product is needed for calibrating IMERG within the YGC region. The GPCC exhibited a significant negative bias in the region to the south of Galongla Mountain due to a lack of precipitation observation data in this region before the establishment of the INVC-RGs. This deficiency contributed to the larger bias in the GPCC data. In contrast, the presence of CMA-RGs in the region to the north of Galongla Mountain improved the quality of the GPCC and IMERG. The INVC program will continue choosing appropriate precipitation observation stations and providing grid-level observation data for the YGC region to enrich observations and improve satellite rainfall estimates.

The results of this study filled gaps in IMERG validation and provided insights into its strengths and weaknesses in complex terrains. The findings could contribute to enhancing satellite-based precipitation estimates and understanding the dynamics of the TP's water cycle.

**Author Contributions:** All authors contributed significantly to this work. Conceptualization, L.L., X.C. and Y.M.; methodology, L.L. and X.C.; software, L.L., X.C. and W.Z.; validation, L.L. and X.C.; formal analysis, L.L., X.C., D.C. and X.X.; investigation, L.L. and X.C.; resources, L.L. and X.C.; data curation, L.L., Y.L., D.C. and X.X.; writing—original draft preparation, L.L.; writing—review and editing, L.L., X.C. and Y.M.; supervision, X.C., H.Z. and Y.M.; project administration, X.C., H.Z. and Y.M; funding acquisition, X.C., H.Z. and Y.M. All authors have read and agreed to the published version of the manuscript.

**Funding:** This research was funded by the Second Tibetan Plateau Scientific Expedition and Research (STEP) Program (grant Nos. 2019QZKK0105, 2019QZKK0103), the National Natural Science Foundation of China (42230610, 41975009, 42275069).

**Data Availability Statement:** The data from INVC-RGs can be downloaded from the website: http:// data.tpdc.ac.cn/zh-hans/disallow/e68f1de1-3a13-4ae1-90e0-9e3a3f57f912/, accessed on 26 February 2023. The data from CMA-RGs can be downloaded from http://www.cma.gov.cn, accessed on 27 January 2023. The IMERG product can be downloaded from https://disc.gsfc.nasa.gov/ accessed on 13 December 2022, and the GPCC product can be downloaded from https://opendata.dwd.de/climate_environment/GPCC/ html/download_gate.html, accessed on 8 November 2022.

**Acknowledgments:** We thank the Foundation for its support and the developers of the operational satellite precipitation products and precipitation gauge measurements. We thank the reviewers and editors for their constructive comments.

**Conflicts of Interest:** The authors declare no conflict of interest.

## Appendix A

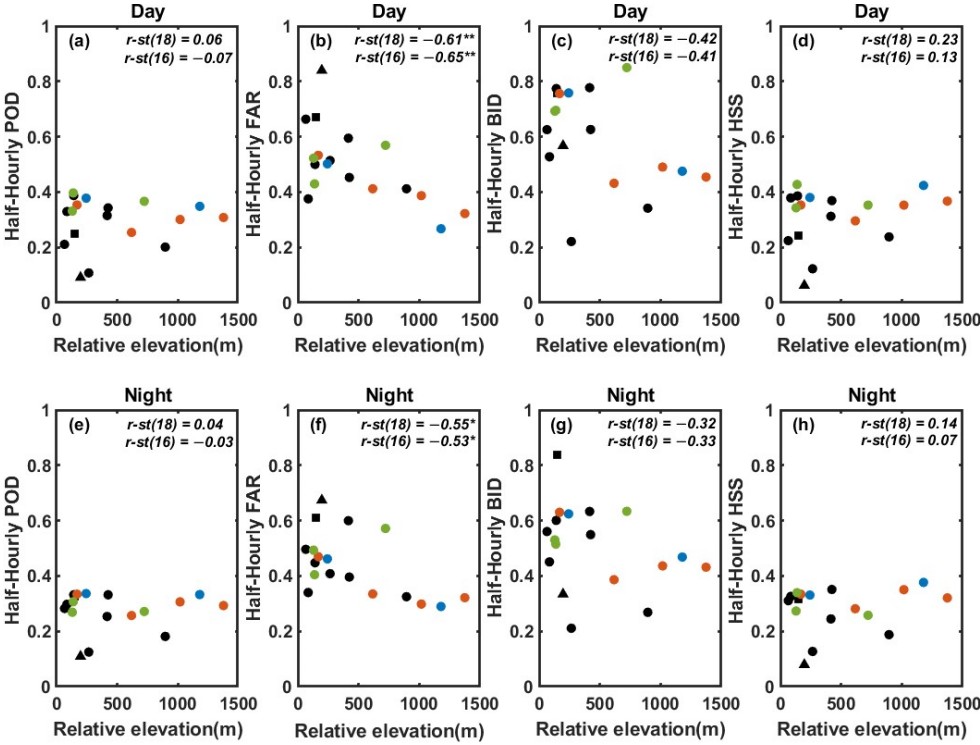

**Figure A1.** Scatter plots of the relative elevations (unit: m) of the sites versus the four scores (POD, FAR, BID, and HSS) at (**a**–**d**) day and (**e**–**h**) night on the half-hourly scales. The circles represent the sites in the GPCC01 grid, the squares represent sites in the GPCC02 grid, and the triangles represent the sites in the GPCC03 grid. The blue circles indicate the IMERG01 grid; the red circles indicate the IMERG02 grid; and the green circles indicate the IMERG03 grid. The correlations (r) between the four scores and the DEMs are noted in the upper right corner of each panel: r-all represents the correlations derived from all 22 sites at the daily scale, r-st18 represents the correlations derived from the 18 non-CMA sites, and r-st16 represents the correlations of the 16 non-CMA sites in the GPCC01 grid. * indicates a *p*-value of 0.05, and ** indicates a *p*-value of 0.01.

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
