# Peer review of "Implications for Validation of IMERG Satellite Precipitation in a Complex Mountainous Region"

_remotesensing, doi:10.3390/rs15184380_

Round 1

Reviewer 1 Report

The study evaluating theIMERG in the complex mountainous region of Yarlung Tsangbo Grand Canyon provides valuable insights into the performance of satellite-based precipitation retrievals. The findings highlight the limitations of IMERG in accurately estimating total rainfall, particularly due to under-detection of rainfall events. The study also investigates the impact of terrain factors on the performance metrics, which further enhances its significance for both users and developers of satellite-based precipitation products. The manuscript presents a comprehensive analysis; however, several points need to be addressed to strengthen the paper.

Major Comments:

(1)The study utilized a relatively short time series of 2-3 years for the precipitation data, which could potentially introduce significant uncertainties in the results. To improve the robustness and reliability of the findings, it is strongly recommended to extend the duration of the precipitation dataset. By incorporating a more extended period of observations, the study would gain enhanced statistical power, better capturing the inter-annual variability and long-term trends in precipitation patterns. Therefore, expanding the temporal coverage of the precipitation data is crucial for providing a more solid foundation for the research outcomes and subsequent implications for users and developers.

(2)The research findings would greatly benefit from a more thorough and quantitative analysis, taking into account various aspects such as temporal and spatial considerations, as well as addressing uncertainties. By incorporating a more detailed and quantitative analysis, encompassing temporal, spatial, and uncertainty considerations, the research outcomes will provide a more rigorous evaluation of IMERG in complex mountainous regions. This comprehensive approach will enable users and developers to make more informed decisions and improve the applicability and reliability of satellite-based precipitation retrievals in such challenging terrains.

(3)The observation that IMERG captures the early morning peak precipitation but underestimates its amplitude raises important questions about the causes of this diurnal variation bias. The authors should explore possible reasons for this bias and suggest potential ways to mitigate it.

Minor Comments:

(1) The manuscript requires careful proofreading for language and clarity. Several sentences are ambiguous and may be confusing for readers. The authors are advised to revise these sentences for better understanding.

(2) The study addressing the evaluation of IMERG in the Yarlung Tsangbo Grand Canyon provides valuable insights into the performance of satellite-based precipitation retrievals in complex mountainous regions. However, in the discussion part, more comparisons with studies from other regions are needed to find out the similarities and differences and possible reasons.

The study evaluating theIMERG in the complex mountainous region of Yarlung Tsangbo Grand Canyon provides valuable insights into the performance of satellite-based precipitation retrievals. The findings highlight the limitations of IMERG in accurately estimating total rainfall, particularly due to under-detection of rainfall events. The study also investigates the impact of terrain factors on the performance metrics, which further enhances its significance for both users and developers of satellite-based precipitation products. The manuscript presents a comprehensive analysis; however, several points need to be addressed to strengthen the paper.

Major Comments:

(1)The study utilized a relatively short time series of 2-3 years for the precipitation data, which could potentially introduce significant uncertainties in the results. To improve the robustness and reliability of the findings, it is strongly recommended to extend the duration of the precipitation dataset. By incorporating a more extended period of observations, the study would gain enhanced statistical power, better capturing the inter-annual variability and long-term trends in precipitation patterns. Therefore, expanding the temporal coverage of the precipitation data is crucial for providing a more solid foundation for the research outcomes and subsequent implications for users and developers.

(2)The research findings would greatly benefit from a more thorough and quantitative analysis, taking into account various aspects such as temporal and spatial considerations, as well as addressing uncertainties. By incorporating a more detailed and quantitative analysis, encompassing temporal, spatial, and uncertainty considerations, the research outcomes will provide a more rigorous evaluation of IMERG in complex mountainous regions. This comprehensive approach will enable users and developers to make more informed decisions and improve the applicability and reliability of satellite-based precipitation retrievals in such challenging terrains.

(3)The observation that IMERG captures the early morning peak precipitation but underestimates its amplitude raises important questions about the causes of this diurnal variation bias. The authors should explore possible reasons for this bias and suggest potential ways to mitigate it.

Minor Comments:

(1) The manuscript requires careful proofreading for language and clarity. Several sentences are ambiguous and may be confusing for readers. The authors are advised to revise these sentences for better understanding.

(2) The study addressing the evaluation of IMERG in the Yarlung Tsangbo Grand Canyon provides valuable insights into the performance of satellite-based precipitation retrievals in complex mountainous regions. However, in the discussion part, more comparisons with studies from other regions are needed to find out the similarities and differences and possible reasons.

Reviewer 2 Report

The study used ground observation data in the YGC region of SETP to evaluate IMERG’s half-hourly and daily scale data on precipitation event detection, rainfall amount estimation and diurnal variation reproduction capabilities. The influences of altitude and topography factors on IMERG’s detection performance are investigated and the product’s uncertainties in the instrument perspective was quantified. The study’s results will help to deeply improve understanding in IMERG’s precipitation data over the YGC region. A Major revision is recommended:

1)Writing in logical structure needs revised. Such as there are seven subtitles under the title “2. Datasets and Methods”. It is not clear what are datasets and what are methods. Suggestion: two subtitles are enough: 2.1 Datasets and 2.2 Methods, categorizing contents in seven subtitles into these two subtitles.

2)”L277 “TI1(site)=DEM(site)-Mean(DEM(IMERG))”. How was the “Mean(DEM(IMERG))‘’ calculated?Is it “Mean DEM of an IMERG grid” shown in Fig.2a. An IMERG grid has a DEM, I do not understand how to get “Mean DEM of an IMERG grid”.

3)L286 TI2(site) means difference between DEM(site) and the average one of the eight grid points surrounding the site. What is the difference between TI2 and TI1? I wonder why these two factors were used to represent terrain factors (The results shown in Fig.5,6 are not good). Suggestion: a site relative elevation can be defined as DEM(site)minus DEM(valley bottom). 

4)L316-317 Are the thresholds for light, moderate, heavy, and extreme precipitation from observation data same for IMERG precipitation?

5) L538-562 stated why a significant negative correlation (52%) between the FAR and TI1 in the case of half-hourly precipitation events. Among these, there is no connection to the valley circulation which seems to be inconsistent with the statement in L529-537. Suggestion to class the Half-Hourly data into Day and Night and have a look at difference between the correlations during Day and Night.

6) In Table 5, the metric NME is got based on whether mean diurnal variation of the precipitation or average NME by each diurnal variation of the precipitation?

7) Fig. 10,11 represents an annual cycle (or seasonal variation) of mean monthly rainfall, so the statement “Monthly Variations” in L763 and other position throughout the text is error.

Minors comments

1)L278 “gague” into “gauge”

1) Which meaning are the subscript 1st, 2nd, 3rd shown in table 1, 3?

2)Too many abbreviations in text, such as L 573 “NUBF”, L 575”SFIs”. What meaning was “PMWs” in L552,554

1) misspelled words , such as 'gague'

2)conceptual error in climate, such as "monthly variation"

Reviewer 3 Report

This manuscript proposes the validation of satellite precipitation product IMERG over the Tibetan Plateau. Validation methods include probability of detection, false alarm ratio, bias in detection and Heidke skill score etc. Although the research method is not novel, it has important contributions to the regional IMERG users and the developers of IMERG. The following are my detailed comments and I hope that those would be helpful to improve the quality of this manuscript.

1.      In section 2, line 216-218 said the extreme values were removed. The criteria for deleting extreme values ​​should be explained. For example, rainfall that is greater than/less than the observed amount is an extreme value.

Are extreme values ​​defined as above 98% of ground-based observation data as stated in Section 2.6 (line 315-317)? If not, clearly state the criteria for deleting extreme value data. If so, you should introduce the definition of extreme value first, and then explain the quality control, which will make it easier for readers to read.

In addition, the amount of raw data and the amount of data after elimination should be clearly stated in this manuscript.

2.      Section 2.3 presents three versions of IMERG that differ only in how soon after observations are available. If so why not use the most recent data (IMERG-E), there should be other reasons for the authors to use IMERG-F in this study, please explain the reasons.

3.      Lines 380-382 of Section 3.1 describe as the elevation increases, the proportion of half-hourly hits decreases, but from Figure 3a, only the five stations with the highest terrain have such a trend, and the others are not obvious, so there is not enough evidence to draw such a conclusion.

4.      Lines 385-386. The POD refers to the proportion of rainfall events detected by IMERG when rainfall is observed by the RGs. Table 2 (in Section 2.7) has already explained it, and there is no need to explain POD again.

5.      Lines 398-400. The YGC area is largely characterized by orographic rainfall, which is influenced by the solar-forced diurnal cycle and multi-scale circulations. This study did not discuss and analyze variables such as solar factors or circulation of any scale. Why is there such a statement?

6.      Section 3.1 describes the diurnal cycle in the YGC area can be generally divided into two categories (Figure 9), could authors mark which stations are north and south of 80K in Figure 9, for the convenience of readers, although there are station names on the figure 9, but the station names in Figure 1 cannot be seen clearly. Perhaps it is not clearly marked, and it is not easy to tell which station is north of 80K from Figure 9, and it is also impossible to judge whether it is single rainfall peaks and multiple rainfall peaks as mentioned in the manuscript.

7.      The results of the Assessment of Precipitation Events in section 3.1 are consistent with the results of similar past studies, but it is not clear where the contribution is in this section? The authors should try to find out the contribution that is different from other studies and increase the value of the manuscript.

8.      The first paragraph of Section 3.2 describes the analysis and discussion of RAINFALL EVENTS CORRECTLY HIT by IMERG. However, the results of section 3.2 (lines 489 - 491) show that the underestimation of IMERG in the YGC area is the result of a HIGH MISSED ALARM RATE for precipitation events and the underestimation of rainfall amounts. Isn't such a conclusion contradictory?

9.      Section 3.3 pointed out that with the increase of altitude, the changes of POD decrease, BID increase, and HSS decrease, these conclusions are similar to other studies and reasonable, but why is this only in the half-hourly scale, and there is no such phenomenon in daily scale?

10.  Lines 614-615 - The hit events in daily precipitation include both half-hour hit and false and missed events, which leads to a larger random bias (Figure 7e). However, the definition of hit in lines 322-323 does not include false and miss events.

The authors of lines 322-326 have their rules for hit, miss, false alarm, and correct negative. The research process and text writing of this manuscript must strictly abide by its rules.

11.  Lines 658-659, h-1 should be superscripted with -1, like h-1.

12.  The rainfall in Figure 9 is all below 1 mm, while it is written as the tipping-bucket RGs with a minimum detection unit of 0.2 mm/tip in lines 208-209. Please explain the source and process of the data in Figure 9, such as rainfall values ​​below 0.2 mm.

13.  The figures in the manuscript should enhance the resolution and enlarge the text in the figures for the convenience of readers.

Reviewer 4 Report

Verifying the validity and correctness of IMERG precipitation products in the Brahmaputra region is of great scientific significance for strengthening climate change research in sparsely populated areas. The manuscript analyzes the factors affecting precipitation from various aspects, and examines the quality of IMERG products by comparing with commonly used precipitation products. However, the results part of the manuscript is too long and the discussion part is too short, so the structure of the manuscript needs to be adjusted. It is recommended that the manuscript be published after minor revisions.

L171 Add the introduction of hydrometeorology.

Round 2

Reviewer 2 Report

The manuscript has been revised based on my comments and suggestions and  can be accepted.

Reviewer 3 Report

The authors have made appropriate improvements and responses to the problems I pointed out, and I recommend accepting them for publication.